# Online Control with Adversarial Disturbance for Continuous-time Linear Systems

**Jingwei Li**
IIIS, Tsinghua University
Shanghai Qizhi Institute
ljw22@mails.tsinghua.edu.cn

**Jing Dong**
The Chinese University of Hong Kong, Shenzhen
jingdong@link.cuhk.edu.cn

**Can Chang**
IIIS, Tsinghua University
cc22@mails.tsinghua.edu.cn

**Baoxiang Wang**
The Chinese University of Hong Kong, Shenzhen
bxiangwang@cuhk.edu.cn

**Jingzhao Zhang**
IIIS, Tsinghua University
Shanghai Qi zhi Institute
jingzhaoz@mail.tsinghua.edu.cn

## Abstract

We study online control for continuous-time linear systems with finite sampling rates, where the objective is to design an online procedure that learns under non-stochastic noise and performs comparably to a fixed optimal linear controller. We present a novel two-level online algorithm, by integrating a higher-level learning strategy and a lower-level feedback control strategy. This method offers a practical and robust solution for online control, which achieves sublinear regret. Our work provides the first nonasymptotic results for controlling continuous-time linear systems with finite number of interactions with the system. Moreover, we examine how to train an agent in domain randomization environments from a non-stochastic control perspective. By applying our method to the SAC (Soft Actor-Critic) algorithm, we achieved improved results in multiple reinforcement learning tasks within domain randomization environments. Our work provides new insights into non-asymptotic analyses of controlling continuous-time systems. Furthermore, our work brings practical intuition into controller learning under non-stochastic environments.

## 1 Introduction

A major challenge in robotics is to deploy simulated controllers into real-world. This process, known as sim-to-real transfer, can be difficult due to misspecified dynamics, unanticipated real-world perturbations, and non-stationary environments. Various strategies have been proposed to address these issues, including domain randomization, meta-learning, and domain adaptation [20, 10, 21]. Although they have shown great effectiveness in experimental results, training agents within these setups poses a significant challenge. To accommodate different environments, the strategies developed by agents tend to be overly conservative [26, 4] or lead to suboptimal outcomes [43, 27].

In this work, we provide an analysis of the sim-to-real transfer problem from an online control perspective. Online control focuses on iteratively updating the controller after deployment (i.e.,

38th Conference on Neural Information Processing Systems (NeurIPS 2024).

online) based on collected trajectories. Significant progress has been made in this field by applying insights from online learning to linear control problems [2, 1, 12, 19, 11, 8, 6, 16].

Following this line of work, we approach the sim-to-real transfer issue for continuous-time linear systems as a non-stochastic control problem, as explored in previous works [19, 11, 8]. These studies provide regret bounds for an online controller that lacks prior knowledge of system perturbations. However, a gap remains as no previous analysis has specifically investigated continuous-time systems, but real world systems often evolve continuously in time.

Existing literature on online continuous control is limited [42, 22, 13, 32]. Most continuous control research emphasizes the development of model-free algorithms, such as policy iteration, under the assumption of noise absence. Recently, [8] examined online continuous-time linear quadratic control problem and achieves sublinear regret. However, it relies on the assumption of standard Brownian noise instead of non-stochastic noise that may not always hold true in real-world applications. This leads us to the crucial question:

*Is it possible to design an online non-stochastic control algorithm*
*in a continuous-time setting that achieves sublinear regret?*

Our work addresses this question by proposing a two-level online controller. The higher-level controller symbolizes the policy learning process and updates the policy at a low frequency to minimize regret. Conversely, the lower-level controller delivers high-frequency feedback control input to reduce discretization error. Our proposed algorithm results in regret bounds for continuous-time linear control in the face of non-stochastic disturbances.

Furthermore, we implement the ideas from our theoretical analysis and test them in several experiments. Note that the key difference between our algorithm and traditional online policy optimization is that we utilize information from past states with some skips to enable faster adaptation to environmental changes. Although the aforementioned concepts are often adopted experimentally as frame stacking and frame skipping, there is relatively little known about the appropriate scenarios for applying these techniques. Our analysis and experiments demonstrate that these techniques are particularly effective in developing adaptive policies for uncertain environments. We choose the task of training agents in a domain randomization environment to evaluate our method, and the results confirm that these techniques substantially improve the agents' performance.

## 2  Related Works

The control theory of linear dynamical systems under disturbances has been thoroughly examined in various contexts, such as linear quadratic stochastic control [7], robust control [37, 23], system identification [17, 24, 9, 25]. However, most of these problems are investigated in non-robust settings, with robust control being the sole exception where adversarial perturbations in the dynamic are permitted. In this scenario, the controller solves for the optimal linear controller in the presence of worst-case noise. Nonetheless, the algorithms designed in this context can be overly conservative as they optimize over the worst-case noise, a scenario that is rare in real-world applications. We will elaborate on the difference between robust control and online non-stochastic control in Section 3.

**Online Control**   There has been a recent surge of interest in online control, as demonstrate by studies such as [2, 1, 12]. In online control, the player interacts with the environment and updates the policy in each round aiming to achieve sublinear regret. In scenarios with stochastic Gaussian noise, [12] provides the first efficient algorithm with an $O(\sqrt{T})$ regret bound. However, in real-world applications, the assumption of Gaussian distribution is often unfulfilled.

[3] pioneers research on non-stochastic online control, where the noises can be adversarial. Under general convex costs, they introduce the Disturbance-Action Policy Class. Using an online convex optimization (OCO) algorithm with memory, they achieve an $O(\sqrt{T})$ regret bound. Subsequent studies extend this approach to other scenarios, such as quadratic costs [8], partial observations [36, 35] or unknown dynamical systems [19, 11]. Other works yield varying theoretical guarantees like online competitive ratio [15, 33].

**Online Continuous Control**   Compared to online control, there has been relatively little research on model-based continuous-time control. Most continuous control works focus on developing model-free

algorithms such as policy iteration (e.g. [42, 22, 32]), typically assuming zero-noise. This is because analyzing the system when transition dynamics are represented by differential equations, rather than recurrence formulas, poses a significant challenge.

Recently, [8] studies online continuous-time linear quadratic control with standard Brownian noise and unknown system dynamics. They propose an algorithm based on the least-square method, which estimates the system's coefficients and solves the corresponding Riccati equation. The papers [34, 14] also focus on online control setups with continuous-time stochastic linear systems and unknown dynamics. They achieve $O(\sqrt{T}\log T)$ regret by different approaches. [34] uses the Thompson sampling algorithm to learn optimal actions. [14] takes a randomized-estimates policy to balance exploration and exploitation. The main difference between [8, 34, 14] and our paper is that they consider stochastic noise of Brownian motion which can be quite stringent and may fail in real-world applications, while the noise in our setup is non-stochastic. This makes our analysis completely different from theirs.

**Domain Randomization**    Domain randomization, which is proposed by [39], is a commonly used technique for training agents to adapt to different (real) environments by training in randomized simulated environments. From the empirical perspective, many previous works focus on designing efficient algorithms for learning in a randomized simulated environment (by randomizing environmental settings, such as friction coefficient) such that the algorithm can adapt well in a new environment, [29, 44, 26, 28, 30]. Other works study how to effectively randomize the simulated environment so that the trained algorithm would generalize well in other environments [43, 27, 38]. However, prior research has not explored how to apply certain theoretical analysis ideas to train agents in domain-randomized environments. Limited previous works, such as [10] and [21], concentrate on theoretically analyzing the sim-to-real gap within specific domain randomization models but they do not test their algorithms in real domain randomization environments.

## 3    Problem Setting

In this paper, we consider the online non-stochastic control for continuous-time linear systems. Therefore, we provide a brief overview below and define our notations.

### 3.1    Continuous-time Linear Systems

The Linear Dynamical System can be considered a specific case of a continuous Markov decision process with linear transition dynamics. The state transitions are governed by the following equation:

$$\dot{x}_t = Ax_t + Bu_t + w_t \,,$$

where $x_t$ is the state at time $t$, $u_t$ is the action taken by the controller at time $t$, and $w_t$ represents the disturbance at time $t$. Follow the setup of [3], we assume $x_0 = 0$. We do not make any strong assumptions about the distribution of $w_t$, and we also assume that the distribution of $w_t$ is unknown to the learner beforehand. This implies that the disturbance sequence $w_t$ can be selected adversarially.

When the action $u_t$ is applied to the state $x_t$, a cost $c_t(x_t, u_t)$ is incurred. Here, we assume that the cost function $c_t$ is convex. However, this cost is not known in advance and is only revealed after the action $u_t$ is implemented at time $t$. In the system described above, an online policy $\pi$ is defined as a function that maps known states to actions, i.e., $u_t = \pi(\{x_\xi | \xi \in [0, t]\})$. Our goal, then, is to design an algorithm that determines such an online policy to minimize the cumulative cost incurred. Specifically, for any algorithm $\mathcal{A}$, the cost incurred over a time horizon $T$ is:

$$J_T(\mathcal{A}) = \int_0^T c_t(x_t, u_t)dt \,.$$

In scenarios where the policy is linear (i.e., a linear controller), such that $u_t = -Kx_t$, we use $J(K)$ to denote the cost of a policy $K \in \mathcal{K}$ from a certain class $\mathcal{K}$.

### 3.2    Difference between Robust and Online Non-stochastic Control

While both robust and online non-stochastic control models incorporate adversarial noise, it's crucial to understand that their objectives differ significantly.

The objective function for robust control, as seen in [37, 23], is defined as:

$$\min_{u_1} \max_{w_{1:T}} \min_{u_2} \ldots \min_{u_t} \max_{w_T} J_T(\mathcal{A}),$$

Meanwhile, the objective function for online non-stochastic control, as discussed in [3], is:

$$\min_{\mathcal{A}} \max_{w_{1:T}} (J_T(\mathcal{A}) - \min_{K \in \mathcal{K}} J_T(K)).$$

Note that the robust control approach seeks to directly minimize the cost function, while online non-stochastic control targets the minimization of regret, which is the discrepancy between the actual cost and the cost associated with a baseline policy. Additionally, in robust control, the noise at each step can depend on the preceding policy, whereas in online non-stochastic control, all the noise is predetermined (though unknown to the player).

### 3.3 Assumptions

We operate under the following assumptions throughout this paper. To be concise, we denote $\|\cdot\|$ as the $L_2$ operator norm of the vector and matrix. Firstly, we make assumptions concerning the system dynamics and noise:

**Assumption 1.** The matrices that govern the dynamics are bounded, meaning $\|A\| \leq \kappa_A$ and $\|B\| \leq \kappa_B$, where $\kappa_A$ and $\kappa_B$ are constants. Moreover, the perturbation and its derivative are both continuous and bounded: $\|w_t\|, \|\dot{w}_t\| \leq W$, with $W$ being a constant.

These assumptions ensure that we can bound the states and actions, as well as their first and second-order derivatives. Next, we make assumptions regarding the cost function:

**Assumption 2.** The costs $c_t(x, u)$ are convex in $x$ and $u$. Additionally, if there exists a constant $D$ such that $\|x\|, \|u\| \leq D$, then we have the following inequalities of the costs: $|c_t(x, u)| \leq \beta D^2, \|\nabla_x c_t(x, u)\|, \|\nabla_u c_t(x, u)\| \leq GD, |c_{t_1}(x, u) - c_{t_2}(x, u)| \leq L|t_1 - t_2|D^2,$

where $\beta, G$ and $L$ are constants corresponding to the cost function. This assumption implies that if the differences between states and actions are small, then the error in their cost will also be relatively small.

### 3.4 Strongly Stable Policy

We next describe our baseline policy class introduced in [12]. Note that the continuous system and the discrete system are different. If we consider the approximation over a relatively small interval $h$, we get

$$x_{t+h} = x_t + \int_{s=t}^{t+h} \dot{x}_s ds = x_t + \int_{s=t}^{t+h} Ax_s + Bu_s + w_s ds$$
$$\approx x_t + h(Ax_t + Bu_t + w_t) = (I + hA)x_t + hBu_t + hw_t.$$

Therefore, if we consider the transition of a discrete system $x_{i+1} = \tilde{A}x_i + \tilde{B}u_i + \tilde{w}_i$, we get the approximation $\tilde{A} \approx I + hA$, $\tilde{B} \approx hB$. Hence, we extend the definition of a strongly stable policy [12, 3] in the discrete system to the continuous system as follows:

**Definition 1.** A linear policy $K$ is $(\kappa, \gamma)$-strongly stable if, for any $h > 0$ that is sufficiently small, there exist matrices $L_h, P$ such that $I + h(A - BK) = PL_hP^{-1}$, with the following two conditions:

1. The norm of $L_h$ is strictly smaller than unity and dependent on $h$, i.e., $\|L_h\| \leq 1 - h\gamma$.

2. The controller and transforming matrices are bounded, i.e., $\|K\| \leq \kappa$ and $\|P\|, \|P^{-1}\| \leq \kappa$.

The above definition ensures the system can be stabilized by a linear controller $K$.

### 3.5 Regret Formulation

To evaluate the designed algorithm, we follow the setup in [12, 3] and use regret, which is defined as the cumulative difference between the cost incurred by the policy of our algorithm and the cost incurred by the best policy in hindsight. Let $\mathcal{K}$ denotes the class of strongly stable linear policies, i.e. $\mathcal{K} = \{K : K \text{ is } (\kappa, \gamma)\text{-strongly stable}\}$. Then we try to minimize the regret of algorithm:

$$\min_{\mathcal{A}} \max_{w_{1:T}} \text{Regret}(\mathcal{A}) = \min_{\mathcal{A}} \max_{w_{1:T}} (J_T(\mathcal{A}) - \min_{K \in \mathcal{K}} J_T(K)).$$

## 4  Algorithm Design

In this section, we outline the design of our algorithm and formally define the concepts involved in deriving our main theorem. We summarize our algorithm design as follows:

First, we discretize the total time period $T$ into smaller intervals of length $h$. We use the information at each point $x_h, x_{2h}, \ldots$ and $u_h, u_{2h}, \ldots$ to approximate the actual cost of each time interval, leveraging the continuity assumption. This process does introduce some discretization errors.

Next, we employ the Disturbance-Action policy (DAC) [3]. This policy selects the action based on the current time step and the estimations of disturbances from several past steps. This policy can approximate the optimal linear policy in hindsight when we choose suitable parameters. However, the optimal policy $K^*$ is unknown, so we cannot directly acquire the optimal choice. To overcome this, we employ the OCO with memory framework [5] to iteratively adjust the DAC policy parameter $M_t$ to approximate the optimal solution $M^*$.

After that, we introduce the concept of the ideal state $y_t$ and ideal action $v_t$ that approximate the actual state $x_t$ and action $u_t$. Note that both the state and policy depend on all DAC policy parameters $M_1, M_2, \ldots, M_t$. Yet, the OCO with memory framework only considers the previous $H$ steps. Therefore, we need to consider ideal state and action. $y_t$ and $v_t$ represent the state the system would reached if it had followed the DAC policy $\{M_{t-H}, \ldots, M_t\}$ at all time steps from $t - H$ to $t$, under the assumption that the state $x_{t-H}$ was $0$.

From all the analysis above, we can decompose the regret as three parts: the discretization error $R_1$, the regret of the OCO with memory $R_2$, and the approximation error between the ideal cost and the actual cost $R_3$.

Then we will formally introduce out method and define all the concepts. In the subsequent discussion, we use shorthand notation to denote the cost, state, control, and disturbance variables $c_{ih}, x_{ih}, u_{ih}$, and $w_{ih}$ as $c_i, x_i, u_i$, and $w_i$, respectively.

First, we need to define the Disturbance-Action Policy Class(DAC) for continuous systems:

**Definition 2.** The Disturbance-Action Policy Class(DAC) is defined as:

$$u_t = -K x_t + \sum_{i=1}^{l} M_t^i \hat{w}_{t-i} \,,$$

where $K$ is a fixed strongly stable policy, $l$ is a parameter that signifies the dimension of the policy class, $M_t = \{M_t^1, \ldots, M_t^l\}$ is the weighting parameter of the disturbance at step $t$, and $\hat{w}_t$ is the estimated disturbance:

$$\hat{w}_t = \frac{x_{t+1} - x_t - h(A x_t + B u_t)}{h} \,. \tag{1}$$

We note that this definition differs from the DAC policy in discrete systems [3] as we utilize the estimation of disturbance over an interval $[t, t+h]$ instead of only the noise in time $t$. It counteracts the second-order residue term of the Taylor expansion of $x_t$ and is also an online policy as it only requires information from the previous state.

Our higher-level controller adopts the OCO with memory framework. A technical challenge lies in balancing the approximation error and OCO regret. To achieve a low approximation error, we desire the policy update interval $H$ to be inversely proportional to the sampling distance $h$. However, this relationship may lead to large OCO regret. To mitigate this issue, we introduce a new parameter $m = \Theta(\frac{1}{h})$, representing the lookahead window. We update the parameter $M_t$ only once every $m$ iterations, further reducing the OCO regret without negatively impacting the approximation error:

$$M_{t+1} = \begin{cases} \Pi_{\mathcal{M}}\left(M_t - \eta \nabla g_t(M)\right) & \text{if } t \bmod m == 0 \,, \\ M_t & \text{otherwise} \,. \end{cases}$$

Where $g_t$ is a function corresponding to the loss function $c_t$ and we will introduce later in Algorithm 1. For notational convenience and to avoid redundancy, we denote $\tilde{M}_{[t/m]} = M_t$. We can then define the ideal state and action. Due to the properties of the OCO with memory structure, we need to consider only the previous $Hm$ states and actions, rather than all states. As a result, we introduce the definition of the ideal state and action. During the interval $t \in [im, (i+1)m - 1]$, the learning policy remains unchanged, so we could define the ideal state and action follow the definition in [3]:

**Definition 3.** The ideal state $y_t$ and action $v_t$ at time $t \in [im, (i+1)m-1]$ are defined as

$$y_t = x_t(\tilde{M}_{i-H}, ..., \tilde{M}_i), v_t = -Ky_t + \sum_{j=1}^{l} M_i^j w_{t-i}.$$

where the notation indicates that we assume the state $x_{t-H}$ is 0 and that we apply the DAC policy $\left(\tilde{M}_{i-H}, \ldots, \tilde{M}_i\right)$ at all time steps from $t - Hm$ to $t$.

We can also define the ideal cost in this interval follow the definition in [3]:

**Definition 4.** The ideal cost function during the interval $t \in [im, (i+1)m-1]$ is defined as follows:

$$f_i\left(\tilde{M}_{i-H}, \ldots, \tilde{M}_i\right) = \sum_{t=im}^{(i+1)m-1} c_t\left(y_t\left(\tilde{M}_{i-H}, \ldots, \tilde{M}_i\right), v_t\left(\tilde{M}_{i-H}, \ldots, \tilde{M}_i\right)\right).$$

With all the concepts presented above, we are now prepared to introduce our algorithm:

---

**Algorithm 1** Continuous two-level online control algorithm

---

**Input:** step size $\eta$, sample distance $h$, policy update parameters $H, m$, parameters $\kappa, \gamma, T$.
Define sample numbers $n = \lceil T/h \rceil$, OCO policy update times $p = \lceil n/m \rceil$.
Define DAC policy update class $\mathcal{M} = \left\{\tilde{M} = \left\{\tilde{M}^1 \ldots \tilde{M}^{Hm}\right\} : \left\|\tilde{M}^i\right\| \leq 2h\kappa^3(1-\gamma)^{i-1}\right\}$.
Initialize $M_0 \in \mathcal{M}$ arbitrarily.
**for** $k = 0, \ldots, p-1$ **do**
    **for** $s = 0, \ldots, m-1$ **do**
        Denote the discretization time $r = km + s$.
        Use the action $u_t = -Kx_r + h\sum_{i=1}^{Hm} \tilde{M}_k^i \hat{w}_{r-i}$ during the time period $t \in [rh, (r+1)h]$.
        Observe the new state $x_{r+1}$ at time $(r+1)h$ and record $\hat{w}_r$ according to Equation (1).
    **end for**
    Define the function $g_k(M) = f_k(M, \ldots, M)$.
    Update OCO policy $\tilde{M}_{k+1} = \Pi_{\mathcal{M}}\left(\tilde{M}_k - \eta\nabla g_k(\tilde{M}_k)\right)$.
**end for**

---

## 5 Main Result

In this section, we present the primary theorem of online continuous control regret analysis:

**Theorem 1.** *Under Assumption 1, 2, a step size of $\eta = \Theta(\sqrt{\frac{m}{Th}})$, and a DAC policy update frequency $m = \Theta(\frac{1}{h})$, Algorithm 1 attains a regret bound of*

$$J_T(\mathcal{A}) - \min_{K \in \mathcal{K}} J_T(K) \leq O(nh(1-h\gamma)^{\frac{H}{h}}) + O(\sqrt{nh}) + O(Th).$$

*With the sampling distance $h = \Theta(\frac{1}{\sqrt{T}})$, and the OCO policy update parameter $H = \Theta(\log(T))$, Algorithm 1 achieves a regret bound of*

$$J_T(\mathcal{A}) - \min_{K \in \mathcal{K}} J_T(K) \leq O\left(\sqrt{T}\log(T)\right).$$

Theorem 1 demonstrates a regret that matches the regret of a discrete system [3]. Despite the analysis of a continuous system differing from that of a discrete system, we can balance discretization error, approximation error, and OCO with memory regret by selecting an appropriate update frequency for the policy. Here, $O(\cdot)$ and $\Theta(\cdot)$ are abbreviations for the polynomial factors of universal constants in the assumption.

While we defer the detailed proof to the appendix, we outline the key ideas and highlight them below.

**Challenge and Proof Sketch** We first explain why we cannot directly apply the methods for discrete nonstochastic control from [3] to our work. To utilize Assumption 2, it is necessary first to establish a union bound over the states. In a discrete-time system, it can be easily proved by applying the dynamics inequality $\|x_{t+1}\| \le a\|x_t\| + b$ (where $a < 1$) and the induction method presented in [3]. However, for a continuous-time system, a different approach is necessary because we only have the differential equation instead of the state recurrence formula.

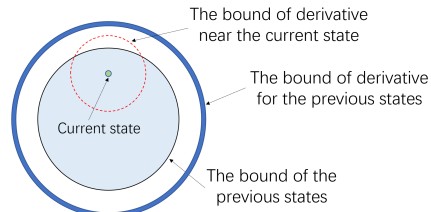

**Figure 1:** Bounding the states and their derivatives separately. We employ Gronwall's inequality with the induction method to bound the states.

To overcome this challenge, we employ Gronwall's inequality to bound the first and second-order derivatives in the neighborhood of the current state. We then use these bounded properties, in conjunction with an estimation of previous noise, to bound the distance to the next state. Through an iterative application of this method, we can argue that all states and actions are bounded.

Another challenge is that we need to discretize the system but we must overcome the curse of dimensionality caused by discretization. In continuous-time systems, the number of states is inversely proportional to the discretization parameter $h$, which also determines the size of the OCO memory buffer. Our regret is primarily composed of three components: the error caused by discretization $R_1$, the regret of OCO with memory $R_2$ and the difference between the actual cost and the approximate cost $R_3$. The discretization error $R_1$ is $O(hT)$, therefore if we achieve $O(\sqrt{T})$ regret, we must choose $h$ no more than $O(\frac{1}{\sqrt{T}})$.

If we update the OCO with memory parameter at each timestep follow the method in [3], we will incur the regret of OCO with memory $R_2 = O(H^{2.5}\sqrt{T})$. The difference between the actual cost and the approximate cost $R_3 = O(T(1 - h\gamma)^H)$. To achieve sublinear regret for the third term, we must choose $H = O(\frac{\log T}{h\gamma})$, but since $h$ is no more than $O(\frac{1}{\sqrt{T}})$, $H$ will be larger than $\Theta(\sqrt{T})$, therefore the second term $R_2$ will definitely exceed $O(\sqrt{T})$.

Therefore, we adjust the frequency of updating the OCO parameters by introducing a new parameter $m$, using a two-level approach and update the OCO parameters once in every $m$ steps. This will incur the third term $R_3 = O(T(1 - h\gamma)^{Hm})$ but keep the OCO with memory regret $R_2 = O(H^{2.5}\sqrt{T})$, so we can choose $H = O(\frac{\log T}{\gamma})$ and $m = O(\frac{1}{h})$. Then the term of $R_2$ is $O(\sqrt{T}\log T)$ and we achieve the same regret compare with the discrete system.

## 6 Experiments

In this section, we apply our theoretical analysis to the practical training of agents. First we highlight the key difference between our algorithm and traditional online policy optimization.

1. *Stack:* While standard online policy optimization learns the optimal policy from the current state $u_t = \phi(x_t)$, an optimal non-stochastic controller employs the DAC policy as outlined in Definition 2. Leveraging information from past states aids the agent in adapting to dynamic environments.

2. *Skip:* Different from the analysis in [3], in a continuous-time system we update the state information every few steps, rather than updating it at every step. This solves the curse of dimensionality caused by discretization in continuous-time system.

The above inspires us with an intuitive strategy for training agents by stacking past observations with some observations to skip. We denote this as *Stack & skip* for convenience. *Stack & skip* is frequently used as a heuristic in reinforcement learning, yet little was known about when and why such a technique could boost agent performance.

How should we evaluate our algorithm in a non-stochastic environment? We opt for learning an optimal policy within a domain randomization environment. In this context, each model's parameters are randomly sampled from a predetermined task distribution. We train policies to optimize performance across various simulated models [41, 29].

We observe that learning in Domain Randomization (DR) significantly differs from stochastic or robust learning problems. In DR, sampling from environmental variables occurs at the beginning of each episode, rather than at every step, distinguishing it from stochastic learning where randomness is step-wise independent and identically distributed. This episodic sampling approach allows agents in DR to exploit environmental conditions and adapt to episodic changes

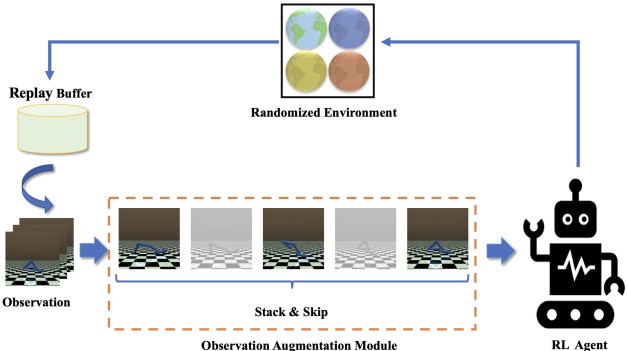

**Figure 2:** Leverage past observation of states with some skip.

within an episode. On the other hand, robust learning focuses on worst-case scenarios depending on an agent's policy. DR, in contrast, is concerned with the distribution of conditions aimed at broad applicability rather than worst-case perturbations.

In the context of non-stochastic control, the disturbance, while not disclosed to the learner beforehand, remains fixed throughout the episode and does not adaptively respond to the control policy. This setup in non-stochastic control shows a clear parallel to domain randomization: fixed yet unknown disturbances in non-stochastic control mirror the unknown training environments in DR. As the agent continually interacts with these environments, it progressively adapts, mirroring the adaptive process observed in domain randomization. Therefore, we propose evaluating our algorithm within a domain randomization training task. Subsequently, we introduce the details of our experimental setup:

**Environment Setting**   We conduct experiments on the hopper, half-cheetah, and walker2d benchmarks using the MuJoCo simulator [40]. The randomized parameters include environmental physical parameters such as damping and friction, as well as the agent properties such as torso size. We set the range of our domain randomization to follow a distribution with default parameters as the mean value, shown in Table 1. When training in the domain randomization environment, the parameter is uniformly sampled from this distribution. To an-

| Environment | Parameters | DR distribution |
|---|---|---|
| Hopper | Joint damping | [0.5, 1.5] |
| | Foot friction | [1, 3] |
| | Height of head | [1.2, 1.7] |
| | Torso size | [0.025, 0.075] |
| Half-Cheetah | Joint damping | [0.005, 0.015] |
| | Foot friction | [3, 7] |
| | Torso size | [0.04, 0.06] |
| Walker2D | Joint damping | [0.05, 0.15] |
| | Density | [500, 1500] |
| | Torso size | [0.025, 0.075] |

**Table 1:** The DR distributions of environment.

alyze the result of generalization, we only change one of the parameters and keep the other parameters as the mean of its distribution in each test environment. We conducted experiments using NVIDIA A40 graphics card.

**Algorithm Design and Baseline**   We design a practical meta-algorithm that converts any standard deep RL algorithm into a domain-adaptive algorithm, shown in Figure 2. In this algorithm, we augment the original state observation $o_t^{\text{old}}$ at time $t$ with past observations, resulting in $o_t^{\text{new}} = [o_t^{\text{old}}, o_{t-m}^{\text{old}}, \ldots, o_{t-(h-1)m}^{\text{old}}]$. Here $h$ is the number of past states we leverage and $m$ is the number of states we skip when we get each of the past states. For clarity in our results, we selected the SAC algorithm for evaluation. We use a variant of Soft Actor-Critic (SAC) [31] and leverage past states with some skip as our algorithm. We compare our algorithm with the standard SAC algorithm training on domain randomization environments as our baseline.

**Impact of Frame Stack and Frame Skip**   To understand the effects of the frame stack number $h$ and frame skip number $m$, we carried out experiments in the hopper environment with different $h$ and $m$. For each parameter we train with 3 random seeds and take the average. Figure 3 shows that the performance increases significantly when the frame stack number is increased from 1 to 3, and

remains roughly unchanged when the frame stack number continues to climb up. Figure 4 shows that the optimal frame skip number is 3, while both too large or too small frame skip numbers result in sub-optimal results. Therefore, in the following experiments we fix the parameter $h = 3, m = 3$. We train our algorithm with this parameter and standard SAC on hopper and test the performance on more environments. Figure 5 shows that our algorithm outperforms the baseline in all environments.

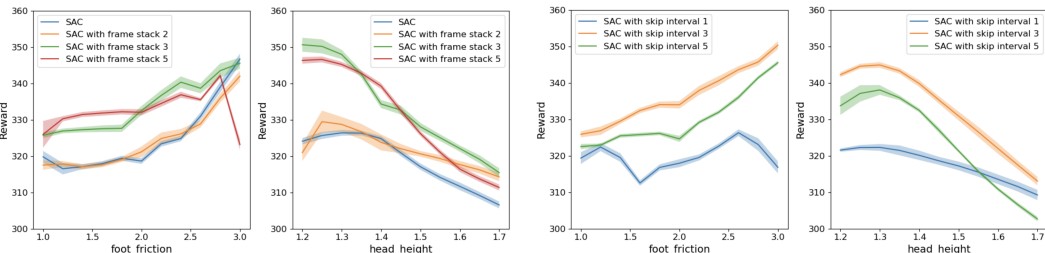

**Figure 3:** Impact of frame stack number.    **Figure 4:** Impact of frame skip number.

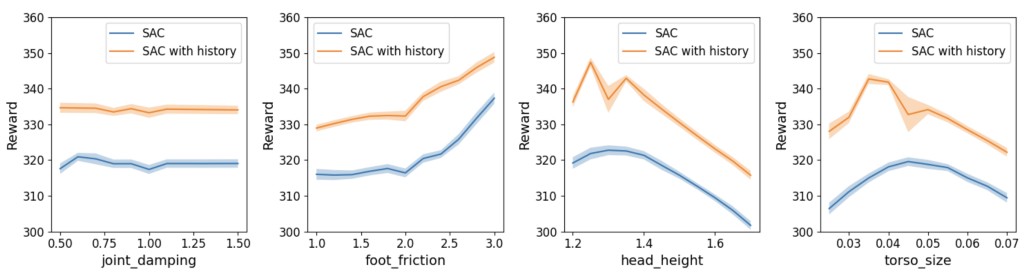

**Figure 5:** Agents' reward in various test environments of hopper.

**Results on Other Environments** Each algorithm was trained using three distinct random seeds in the half-cheetah and walker2d domain randomization (DR) environments. Consistent with previous experiments, we employed a frame stack number of $h = 3$ and frame skip number of $m = 3$. The comparative performance of our algorithm and the baseline algorithm, across various domain parameters, is presented in Figure 6. The result clearly demonstrates that our algorithm consistently outperforms the baseline in all evaluated test environments.

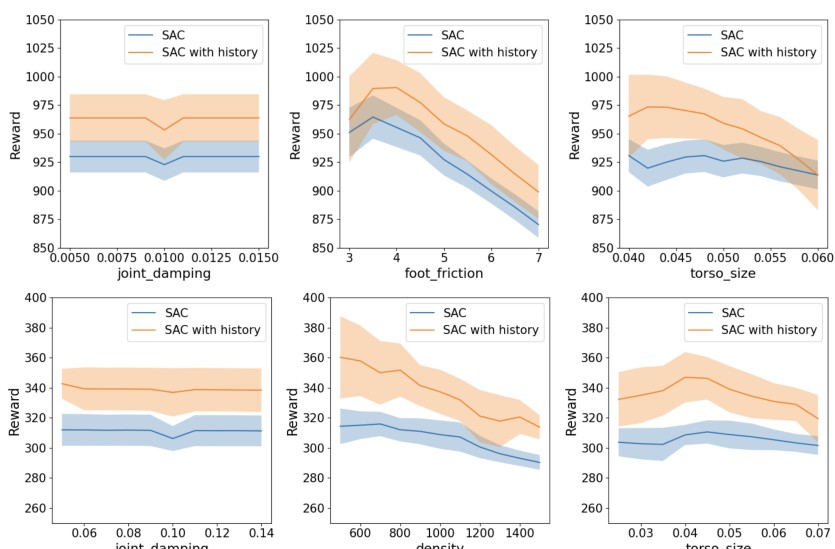

**Figure 6:** Performance in half-cheetah(Top) and walker2d(Bottom).

# 7 Conclusion, Limitations and Future Directions

In this paper, we propose a two-level online controller for continuous-time linear systems with adversarial disturbances, aiming to achieve sublinear regret. This approach is grounded in our examination of agent training in domain randomization environments from an online control perspective. At the higher level, our controller employs the Online Convex Optimization (OCO) with memory framework to update policies at a low frequency, thus reducing regret. The lower level uses the DAC policy to align the system's actual state more closely with the idealized setting.

In our empirical evaluation, applying our algorithm's core principles to the SAC (Soft Actor-Critic) algorithm led to significantly improved results in multiple reinforcement learning tasks within domain randomization environments. This highlights the adaptability and effectiveness of our approach in practical scenarios.

It is important to note that our theoretical analysis depends on the known dynamics of the system and the assumption of convex costs. This reliance could represent a limitation to our method, as it may not adequately address scenarios where these conditions do not hold or where system dynamics are incompletely understood. For future research, there are several promising directions in online non-stochastic control of continuous-time systems. These include extending our methods to systems with unknown dynamics, exploring the impact of assuming strong convexity in cost functions, and shifting the focus from regret to the competitive ratio. Further research can also explore how to utilize historical information more effectively to enhance agent training in domain randomization environments. This might involve employing time series analysis instead of simply incorporating parameters into neural network training.

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

In the appendix we define $n$ as the smallest integer greater than or equal to $\frac{T}{h}$, and we use the shorthand $c_{ih}$, $x_{ih}$, $u_{ih}$, and $w_{ih}$ as $c_i$, $x_i$, $u_i$, and $w_i$, respectively. First we provide the proof of our main theorem there.

## A   Proof of Theorem 1

**Theorem 1.** *Under Assumption 1, 2, a step size of $\eta = \Theta(\sqrt{\frac{m}{Th}})$, and a DAC policy update frequency $m = \Theta(\frac{1}{h})$, Algorithm 1 attains a regret bound of*

$$J_T(\mathcal{A}) - \min_{K \in \mathcal{K}} J_T(K) \leq O(nh(1 - h\gamma)^{\frac{H}{h}}) + O(\sqrt{nh}) + O(Th).$$

*With the sampling distance $h = \Theta(\frac{1}{\sqrt{T}})$, and the OCO policy update parameter $H = \Theta(\log(T))$, Algorithm 1 achieves a regret bound of*

$$J_T(\mathcal{A}) - \min_{K \in \mathcal{K}} J_T(K) \leq O\left(\sqrt{T} \log(T)\right).$$

*Proof.* We denote $u_t^* = K^* x_t^*$ as the optimal state and action that follows the policy specified by $K^*$, where $K^* = \arg\max_{K \in \mathcal{K}} J_T(K)$.

We then discretize and decompose the regret as follows:

$$\begin{aligned}
J_T(\mathcal{A}) - \min_{K \in \mathcal{K}} J_T(K) &= \int_0^T c_t(x_t, u_t) dt - \int_0^T c_t(x_t^*, u_t^*) dt \\
&= \sum_{i=0}^{n-1} \int_{ih}^{(i+1)h} c_t(x_t, u_t) dt - \sum_{i=0}^{n-1} \int_{ih}^{(i+1)h} c_t(x_t^*, u_t^*) dt \\
&= h\left(\sum_{i=0}^{n-1} c_i(x_i, u_i) - \sum_{i=0}^{n-1} c_i(x_i^*, u_i^*)\right) + R_0,
\end{aligned}$$

where $R_0$ represents the discretization error.

We define $p$ as the smallest integer greater than or equal to $\frac{n}{m}$, then the first term can be further decomposed as

$$\begin{aligned}
&\sum_{i=0}^{n-1} c_i(x_i, u_i) - \sum_{i=0}^{n-1} c_i(x_i^*, u_i^*) \\
=& \sum_{i=0}^{p-1} \sum_{j=im}^{(i+1)m-1} c_i(x_i, u_i) - \sum_{i=0}^{p-1} \sum_{j=im}^{(i+1)m-1} c_i(x_i^*, u_i^*) \\
=& \sum_{i=0}^{p-1} \left( \sum_{j=im}^{(i+1)m-1} c_i(x_i, u_i) - \sum_{j=im}^{(i+1)m-1} c_i(y_i, v_i) \right) + \sum_{i=0}^{p-1} \sum_{j=im}^{(i+1)m-1} c_i(y_i, v_i) - \sum_{i=0}^{p-1} \sum_{j=im}^{(i+1)m-1} c_i(x_i^*, u_i^*) \\
=& \sum_{i=0}^{p-1} \left( \sum_{j=im}^{(i+1)m-1} c_i(x_i, u_i) - f_i(\tilde{M}_{i-H}, \ldots, \tilde{M}_i) \right) + \sum_{i=0}^{p-1} f_i(\tilde{M}_{i-H}, \ldots, \tilde{M}_i) \\
&- \min_{M \in \mathcal{M}} \sum_{i=0}^{p-1} f_i(M, \ldots, M) + \min_{M \in \mathcal{M}} \sum_{i=0}^{p-1} f_i(M, \ldots, M) - \sum_{i=0}^{p-1} \sum_{j=im}^{(i+1)m-1} c_i(x_i^*, u_i^*),
\end{aligned}$$

where the last equality is by the definition of the idealized cost function.

Let us denote

$$R_1 = \sum_{i=0}^{p-1} \left( \sum_{j=im}^{(i+1)m-1} c_i(x_i, u_i) - f_i(\tilde{M}_{i-H}, \ldots, \tilde{M}_i) \right),$$

$$R_2 = \sum_{i=0}^{p-1} f_i(\tilde{M}_{i-H}, \ldots, \tilde{M}_i) - \min_{M \in \mathcal{M}} \sum_{i=0}^{p-1} f_i(M, \ldots, M),$$

$$R_3 = \min_{M \in \mathcal{M}} \sum_{i=0}^{p-1} f_i(M, \ldots, M) - \sum_{i=0}^{p-1} \sum_{j=im}^{(i+1)m-1} c_i(x_i^*, u_i^*).$$

Then we have the regret decomposition as

$$\text{Regret}(T) = h(R_1 + R_2 + R_3) + O(hT).$$

We then separately upper bound each of the four terms.

The term $R_0$ represents the error caused by discretization, which decreases as the number of sampling points increases and the sampling distance $h$ decreases. This is because more sampling points make our approximation of the continuous system more accurate. Using Lemma 3, we get the following upper bound: $R_0 \leq O(hT)$.

The term $R_1$ represents the difference between the actual cost and the approximate cost. For a fixed $h$, this error decreases as the number of sample points looked ahead $m$ increases, while it increases as the sampling distance $h$ decreases. This is because the closer adjacent points are, the slower the convergence after approximation. By Lemma 4 we can bound it as $R_1 \leq O(n(1 - h\gamma)^{Hm})$.

The term $R_2$ is incurred due to the regret of the OCO with memory algorithm. Note that this term is determined by learning rate $\eta$ and the policy update frequency $m$. Choosing suitable parameters and using Lemma 5, we can obtain the following upper bound: $R_2 \leq O(\sqrt{n/h})$.

The term $R_3$ represents the difference between the ideal optimal cost and the actual optimal cost. Since the accuracy of the DAC policy approximation of the optimal policy depends on its degree of freedom $l$, a higher degree of freedom leads to a more accurate approximation of the optimal policy. We use Lemma 6 and choose $l = Hm$ to bound this error: $R_3 \leq O(n(1 - h\gamma)^{Hm})$.

By summing up these four terms and taking $m = \Theta(\frac{1}{h})$, we get:

$$\text{Regret}(T) \leq O(nh(1 - h\gamma)^{\frac{H}{h}}) + O(\sqrt{nh}) + O(hT).$$

Finally, we choose $h = \Theta\left(\frac{1}{\sqrt{T}}\right)$, $m = \Theta\left(\frac{1}{h}\right)$, $H = \Theta(log(T))$, the regret is bounded by

$$\text{Regret}(T) \leq O(\sqrt{T} \log(T)).$$

$\square$

## B  Key Lemmas

In this section, we will primarily discuss the rationale behind the proof of our key lemmas. First, we need to prove all the states and actions are bounded.

**Lemma 2.** *Under Assumption 1 and 2, choosing arbitrary $h$ in the interval $[0, h_0]$ where $h_0$ is a constant only depends on the parameters in the assumption, we have for any $t$ and policy $M_i$, $\|x_t\|, \|y_t\|, \|u_t\|, \|v_t\| \leq D$, $\|\dot{x}_t\| \leq D$, $\|x_t - y_t\|, \|u_t - v_t\| \leq \kappa^2(1 + \kappa)(1 - h\gamma)^{Hm+1}D$. In particular, taking all the $M_t = 0$ and $K = K^*$, we can also obtain the inequality of the optimal solution: $\|x_t^*\|, \|u_t^*\| \leq D$.*

The proof of this Lemma mainly use the Gronwall inequality and the induction method. Then we analyze the discretization error of the system.

**Lemma 3.** *Under Assumption 2, Algorithm 1 attains the following bound of $R_0$:*

$$R_0 = \sum_{i=0}^{n-1} \int_{ih}^{(i+1)h} (c_t(x_t, u_t) - c_t(x_t^*, u_t^*))dt - h\sum_{i=0}^{n-1} (c_i(x_i, u_i) - c_i(x_i^*, u_i^*)) \leq (G+L)D^2 hT.$$

This lemma indicates that the discretization error is directly proportional to the sample distance $h$. In other words, increasing the number of sampling points leads to more accurate estimation of system.

Then we analysis the difference between ideal cost and actual cost. The following lemma describes the upper bound of the error by approximating the ideal state and action:

**Lemma 4.** *Under Assumption 1 and 2, Algorithm 1 attains the following bound of $R_1$:*

$$R_1 = \sum_{i=0}^{p-1} \left( \sum_{j=im}^{(i+1)m-1} c_i(x_i, u_i) - f_i\left(\tilde{M}_{i-H}, \ldots, \tilde{M}_i\right) \right) \leq nGD^2\kappa^2(1+\kappa)(1-h\gamma)^{Hm+1}.$$

From this lemma, it is evident that for a fixed sample distance $h$, the error diminishes as the number of sample points looked ahead $m$ increases. However, as the sampling distance $h$ decreases, the convergence rate of this term becomes slower. Therefore, it is not possible to select an arbitrarily small value for $h$ in order to minimize the discretization error $R_0$.

We need to demonstrate that the discrepancy between $x_t$ and $y_t$, as well as $u_t$ and $v_t$, is sufficiently small, given assumption 1. This can be proven by analyzing the state evolution under the DAC policy.

By utilizing Assumption 2 and Lemma 2, we can deduce the following inequality:

$$|c_t(x_t, u_t) - c_t(y_t, v_t)| \leq |c_t(x_t, u_t) - c_t(y_t, u_t)| + |c_t(y_t, u_t) - c_t(y_t, v_t)|$$
$$\leq GD\|x_t - y_t\| + GD\|u_t - v_t\|.$$

Summing over all the terms and use Lemma 2, we can derive an upper bound for $R_1$.

Next, we analyze the regret of Online Convex Optimization (OCO) with a memory term. To analyze OCO with a memory term, we provide an overview of the framework established by [5] in online convex optimization. The framework considers a scenario where, at each time step $t$, an online player selects a point $x_t$ from a set $\mathcal{K} \subset \mathbb{R}^d$. At each time step, a loss function $f_t : \mathcal{K}^{H+1} \to \mathbb{R}$ is revealed, and the player incurs a loss of $f_t(x_{t-H}, \ldots, x_t)$. The objective is to minimize the policy regret, which is defined as

$$\text{PolicyRegret} = \sum_{t=H}^{T} f_t(x_{t-H}, \ldots, x_t) - \min_{x \in \mathcal{K}} \sum_{t=H}^{T} f_t(x, \ldots, x).$$

In this setup, the first term corresponds to the DAC policy we choose, while the second term is used to approximate the optimal strongly stable linear policy.

**Lemma 5.** *Under Assumption 1 and 2, choosing $m = \frac{C}{h}$ and $\eta = \Theta(\frac{m}{Th})$, Algorithm 1 attains the following bound of $R_2$:*

$$R_2 = \sum_{i=0}^{p-1} f_i(\tilde{M}_{i-H}, \ldots, \tilde{M}_i) - \min_{M \in \mathcal{M}} \sum_{i=0}^{p-1} f_i(M, \ldots, M)$$

$$\leq \frac{4a}{\gamma} \sqrt{\frac{GDC^2\kappa^2(\kappa+1)W_0\kappa_B}{\gamma}\left(\frac{GDC\kappa^2(\kappa+1)W_0\kappa_B}{\gamma} + C^2\kappa^3\kappa_B W_0 H^2\right)\frac{n}{h}}.$$

To analyze this term, we can transform the problem into an online convex optimization with memory and utilize existing results presented by [5] for it. By applying their results, we can derive the following bound:

$$\sum_{t=H}^{T} f_t(x_{t-H}, \ldots, x_t) - \min_{x \in \mathcal{K}} \sum_{t=H}^{T} f_t(x, \ldots, x) \leq O\left(D\sqrt{G_f(G_f + LH^2)T}\right).$$

Taking into account the bounds on the diameter, Lipschitz constant, and the gradient, we can ultimately derive an upper bound for $R_2$.

Lastly, we aim to establish a bound on the approximation error between the optimal DAC policy and the unknown optimal linear policy.

**Lemma 6.** *Under Assumption 1 and 2, Algorithm 1 attains the following bound of $R_3$:*

$$R_3 = \min_{M \in \mathcal{M}} \sum_{i=0}^{p-1} f_i(M, ..., M) - \sum_{i=0}^{p-1} \sum_{j=im}^{(i+1)m-1} c_i(x_i^*, u_i^*) \leq 3n(1 - h\gamma)^{Hm} GDW_0 \kappa^3 a(lh\kappa_B + 1).$$

The intuition behind this lemma is that the evolution of states leads to an approximation of the optimal linear policy in hindsight, where $u_t^* = -K^* x_t$ if we choose $M^* = \{M^i\}$, where $M^i = (K - K^*)(I + h(A - BK^*))^i$. Although the optimal policy $K^*$ is unknown, such an upper bound is attainable because the left-hand side represents the minimum of $M \in \mathcal{M}$.

## C  The evolution of the state

In this section we will prove that using the DAC policy, the states and actions are uniformly bounded. The difference between ideal and actual states and the difference between ideal and actual action is very small.

We begin with expressions of the state evolution using DAC policy:

**Lemma 7.** *We have the evolution of the state and action:*

$$x_{t+1} = Q_h^{l+1} x_{t-l} + h \sum_{i=0}^{2l} \Psi_{t,i} \hat{w}_{t-i},$$

$$y_{t+1} = h \sum_{i=0}^{2Hm} \Psi_{t,i} \hat{w}_{t-i},$$

$$v_t = -K y_t + h \sum_{j=1}^{Hm} M_t^j \hat{w}_{t-j}.$$

*where $\Psi_{t,i}$ represent the coefficients of $\hat{w}_{t-i}$:*

$$\Psi_{t,i} = Q_h^i \mathbf{1}_{i \leq l} + h \sum_{j=0}^{l} Q_h^j B M_{t-j}^{i-j} \mathbf{1}_{i-j \in [1,l]}.$$

*Proof.* Define $Q_h = I + h(A - BK)$. Using the Taylor expansion of $x_t$ and denoting $r_t$ as the second-order residue term, we have

$$x_{t+1} = x_t + h\dot{x}_t + h^2 r_t = x_t + h(Ax_t + Bu_t + w_t) + h^2 r_t.$$

Then we calculate the difference between $w_i$ and $\hat{w}_i$:

$$\hat{w}_t - w_t = \frac{x_{t+1} - x_t - h(Ax_t + Bu_t + w_t)}{h} = hr_t.$$

Using the definition of DAC policy and the difference between disturbance, we have

$$
\begin{aligned}
x_{t+1} &= x_t + h\left(Ax_t + B\left(-Kx_t + h\sum_{i=1}^{l} M_t^i \hat{w}_{t-i}\right) + \hat{w}_t - hr_t\right) + h^2 r_t \\
&= (I + h(A - BK))x_t + h\left(Bh\sum_{i=1}^{l} M_t^i \hat{w}_{t-i} + \hat{w}_t\right) \\
&= Q_h x_t + h\left(Bh\sum_{i=1}^{l} M_t^i \hat{w}_{t-i} + \hat{w}_t\right) \\
&= Q_h^2 x_{t-1} + h\left(Q_h\left(Bh\sum_{i=1}^{l} M_{t-1}^i \hat{w}_{t-1-i} + \hat{w}_{t-1}\right)\right) + h\left(Bh\sum_{i=1}^{l} M_t^i \hat{w}_{t-i} + \hat{w}_t\right) \\
&= Q_h^{l+1} x_{t-l} + h\sum_{i=0}^{2l} \Psi_{t,i} \hat{w}_{t-i},
\end{aligned}
$$

where the last equality is by recursion and $\Psi_{t,i}$ represent the coefficients of $\hat{w}_{t-i}$.

Then we calculate the coefficients of $w_{t-i}$ and get the following result:

$$
\Psi_{t,i} = Q_h^i \mathbf{1}_{i \leq l} + h\sum_{j=0}^{l} Q_h^j B M_{t-j}^{i-j} \mathbf{1}_{i-j \in [1,l]}.
$$

By the ideal definition of $y_{t+1}$ and $v_t$ (only consider the effect of the past $Hm$ steps while planning, assume $x_{t-Hm} = 0$), taking $l = Hm$ we have

$$
y_{t+1} = h\sum_{i=0}^{2Hm} \Psi_{t,i} \hat{w}_{t-i},
$$

$$
v_t = -Ky_t + h\sum_{j=1}^{Hm} M_t^j \hat{w}_{t-j}. \qquad \square
$$

Then we prove the norm of the transition matrix is bounded.

**Lemma 8.** *We have the following bound of the transition matrix:*

$$
\|\Psi_{t,i}\| \leq a(lh\kappa_B + 1)\kappa^2(1 - h\gamma)^{i-1}.
$$

*Proof.* By the definition of strongly stable policy, we know

$$
\|Q_h^i\| = \|(PL_h P^{-1})^i\| = \|P(L_h)^i P^{-1}\| \leq \|P\|\|L_h\|^i\|P^{-1}\| \leq a\kappa^2(1 - h\gamma)^i. \qquad (2)
$$

By the definition of $\Psi_{t,i}$, we have

$$
\begin{aligned}
\|\Psi_{t,i}\| &= \left\|Q_h^i \mathbf{1}_{i \leq l} + h\sum_{j=0}^{l} Q_h^j B M_{t-j}^{i-j} \mathbf{1}_{i-j \in [1,l]}\right\| \\
&\leq \kappa^2(1 - h\gamma)^i + ah\sum_{j=1}^{l} \kappa_B \kappa^2(1 - h\gamma)^j(1 - h\gamma)^{i-j-1} \\
&\leq \kappa^2(1 - h\gamma)^i + alh\kappa_B \kappa^2(1 - h\gamma)^{i-1} \leq a(lh\kappa_B + 1)\kappa^2(1 - h\gamma)^{i-1},
\end{aligned}
$$

where the first inequality is due to equation 2, assumption 1 and the condition of $\|M_t^i\| \leq a(1 - h\gamma)^{i-1}$. $\qquad \square$

After that, we can uniformly bound the state $x_t$ and its first and second-order derivative.

**Lemma 9.** *For any $t \in [0, T]$, choosing arbitrary $h$ in the interval $[0, h_0]$ where $h_0$ is a constant only depends on the parameters in the assumption, we have $\|x_t\| \leq D_1$, $\|\dot{x}_t\| \leq D_2$, $\|\ddot{x}_t\| \leq D_3$ and the estimatation of disturbance is bounded by $\|\hat{w}_t\| \leq W_0$. Moreover, $D_1$, $D_2$, $D_3$ are only depend on the parameters in the assumption.*

*Proof.* We prove this lemma by induction. When $t = 0$, it is clear that $x_0$ satisfies this condition. Suppose $x_t \leq D_1$, $\dot{x}_t \leq D_2$, $\ddot{x}_t \leq D_3$, $\hat{w}_t \leq W_0$ for any $t \leq t_0$, where $t_0 = kh$ is the $k$-th discretization point. Then for $t \in [t_0, t_0 + h]$, we first prove that $\dot{x}_t \leq D_2$, $\ddot{x}_t \leq D_3$.

By Assumption 1 and our definition of $u_t$, we know that for any $t \in [t_0, t_0 + h]$. Thus, we have

$$\|\dot{x}_t\| = \|Ax_t + Bu_t + w_t\|$$

$$= \|Ax_t + B(-Kx_{t_0} + h\sum_{i=1}^{l} M_k^i \hat{w}_{k-i}) + w_t\|$$

$$\leq \kappa_A \|x_t\| + \kappa_B \kappa \|x_{t_0}\| + h\sum_{i=1}^{l}(1 - h\gamma)^{i-1}W_0 + W$$

$$\leq \kappa_A \|x_t\| + \kappa_B \kappa D_1 + \frac{W_0}{\gamma} + W \,,$$

where the first inequality is by the induction hypothesis $\hat{w}_t \leq W_0$ for any $t \leq t_0$ and $M_k^i \leq (1 - h\gamma)^{i-1}$, the second inequality is by the induction hypothesis $x_t \leq D_1$ for any $t \leq t_0$.

For any $t \in [t_0, t_0 + h]$, because we choose the fixed policy $u_t \equiv u_{t_0}$, so we have $\dot{u}_t = 0$ and

$$\|\ddot{x}_t\| = \|A\dot{x}_t + B\dot{u}_t + \dot{w}_t\| = \|A\dot{x}_t + \dot{w}_t\| \leq \kappa_A \|\dot{x}_t\| + W \,.$$

By the Newton-Leibniz formula, we have for any $\zeta \in [0, h]$,

$$\dot{x}_{t_0+\zeta} - \dot{x}_{t_0} = \int_0^\zeta \ddot{x}_{t_0+\xi} d\xi \,.$$

Then we have

$$\|\dot{x}_{t_0+\zeta}\| \leq \|\dot{x}_{t_0}\| + \int_0^\zeta \|\ddot{x}_{t_0+\xi}\| d\xi$$

$$\leq \|\dot{x}_{t_0}\| + \int_0^\zeta (\kappa_A \|\dot{x}_{t_0+\xi}\| + W) d\xi$$

$$= \|\dot{x}_{t_0}\| + W\zeta + \kappa_A \int_0^\zeta \|\dot{x}_{t_0+\xi}\| d\xi \,.$$

By Gronwall inequality, we have

$$\|\dot{x}_{t_0+\zeta}\| \leq \|\dot{x}_{t_0}\| + W\zeta + \int_0^\zeta (\|\dot{x}_{t_0}\| + W\xi) \exp(\kappa_A(\zeta - \xi)) d\xi \,.$$

Then we have

$$\|\dot{x}_{t_0+\zeta}\| \leq \|\dot{x}_{t_0}\| + W\zeta + \int_0^\zeta (\|\dot{x}_{t_0}\| + W\zeta) \exp(\kappa_A\zeta)) d\xi$$

$$= (\|\dot{x}_{t_0}\| + W\zeta)(1 + \zeta \exp(\kappa_A\zeta))$$

$$\leq \left(\kappa_A \|x_{t_0}\| + \kappa_B \kappa D_1 + \frac{W_0}{\gamma} + W + Wh\right)(1 + h\exp(\kappa_A h))$$

$$\leq \left((\kappa_A + \kappa_B \kappa)D_1 + \frac{W_0}{\gamma} + W + Wh\right)(1 + h\exp(\kappa_A h))$$

$$\leq \left((\kappa_A + \kappa_B \kappa)D_1 + \frac{W_0}{\gamma} + 2W\right)(1 + \exp(\kappa_A)) \,,$$

where the first inequality is by the relation $\xi \leq \zeta$, the second inequality is by the relation $\zeta \leq h$ and the bounding property of first-order derivative, the third inequality is by the induction hypothesis and the last inequality is due to $h \leq 1$.

By the relation $\|\ddot{x}_t\| \leq \kappa_A\|\dot{x}_t\| + W$, we have
$$\|\ddot{x}_{t_0+\zeta}\| \leq \kappa_A D_2 + W.$$
So we choose $D_3 = \kappa_A D_2 + W$. By the equation, we have

$$
\begin{aligned}
\|\hat{w}_t - w_t\| &= \left\| \frac{x_{t+1} - x_t - h(Ax_t + Bu_t + w_t)}{h} \right\| \\
&= \left\| \frac{x_{t+1} - x_t - h\dot{x}_t}{h} \right\| = \left\| \frac{\int_0^h (\dot{x}_{t+\xi} - \dot{x}_t)d\xi}{h} \right\| = \left\| \frac{\int_0^h \int_0^\xi \ddot{x}_{t+\zeta} d\zeta d\xi}{h} \right\| \\
&\leq \frac{\int_0^h \int_0^\xi \|\ddot{x}_{t+\zeta}\| d\zeta d\xi}{h} \\
&\leq hD_3,
\end{aligned}
$$

where in the second line we use the Newton-Leibniz formula, the inequality is by the conclusion $\|\ddot{x}_t\| \leq D_3$ which we have proved before. By Assumption 1, we have
$$\|\hat{w}_t\| \leq W + hD_3.$$

Choosing $D_3 = \kappa_A D_2 + W$, $W_0 = W + hD_3 = W + h(\kappa_A D_2 + W)$, we get

$$
\begin{aligned}
\|\dot{x}_{t_0+\zeta}\| &\leq ((\kappa_A + \kappa_B\kappa)D_1 + \frac{W_0}{\gamma} + 2W)(1 + \exp(\kappa_A)) \\
&\leq ((\kappa_A + \kappa_B\kappa)D_1 + \frac{W + h(\kappa_A D_2 + W)}{\gamma} + 2W)(1 + \exp(\kappa_A)) \\
&\leq D_2 \left( \frac{h\kappa_A}{\gamma}(1 + \exp(\kappa_A)) \right) + \left( (\kappa_A + \kappa_B\kappa)D_1 + \frac{(1 + h + 2\gamma)W}{\gamma} \right)(1 + \exp(\kappa_A)).
\end{aligned}
$$

Using the notation

$$
\begin{aligned}
\beta_1 &= \frac{h\kappa_A}{\gamma}(1 + \exp(\kappa_A)), \\
\beta_2 &= \left( (\kappa_A + \kappa_B\kappa)D_1 + \frac{2(1 + \gamma)W}{\gamma} \right)(1 + \exp(\kappa_A)).
\end{aligned}
$$

When $h < \frac{\gamma}{2\kappa_A(1+\exp(\kappa_A))}$, we have $\beta_1 < \frac{1}{2}$. Taking $D_2 = 2\beta_2$ we get
$$\|\dot{x}_{t_0+\zeta}\| \leq \beta_1 D_2 + \beta_2 \leq D_2.$$

So we have proved that for any $t \in [t_0, t_0 + h]$, $\|\dot{x}_t\| \leq D_2$, $\|\ddot{x}_t\| \leq D_3$, $\|\hat{w}_t\| \leq W_0$.

Then we choose suitable $D_1$ and prove that for any $t \in [t_0, t_0 + h]$, $\|x_t\| \leq D_1$.

Using Lemma 7, we have
$$x_{t+1} = h \sum_{i=0}^t \Psi_{t,i} \hat{w}_{t-i}.$$

By the induction hypothesis of bounded state and estimation noise in $[0, t_0]$ together with Lemma 8, we have

$$
\begin{aligned}
\|x_{t+1}\| &\leq h \sum_{i=0}^t (lh\kappa_B + 1)\kappa^2 (1 - h\gamma)^i (W + hD_3) \\
&\leq \frac{(lh\kappa_B + 1)\kappa^2 (W + hD_3)}{\gamma}.
\end{aligned}
$$

Then, by the Taylor expansion and the inequality $\dot{x}_t \leq D_2$ , we have for any $\zeta \in [0, h]$,

$$\|x_{t+1} - x_{t+\zeta}\| = \|\int_\zeta^h \dot{x}_{t+\xi}d\xi\| \leq (h - \zeta)D_2 \leq hD_2 \,.$$

Therefore we have

$$\|x_{t+\zeta}\| \leq \|x_{t+1}\| + hD_2 \leq \frac{(lh\kappa_B + 1)\kappa^2(W + hD_3)}{\gamma} + hD_2$$

$$= \frac{(lh\kappa_B + 1)\kappa^2 W(1 + h)}{\gamma} + hD_2 \left(\frac{(lh\kappa_B + 1)\kappa^2\kappa_A}{\gamma} + 1\right)$$

$$\leq \frac{(l\kappa_B + 1)2\kappa^2 W}{\gamma} + hD_2 \left(\frac{(l\kappa_B + 1)\kappa^2\kappa_A}{\gamma} + 1\right) \,.$$

In the last inequality we use $h \leq 1$.

By the relation $D_2 = \beta_2/(1 - \beta_1)$ and $\beta_1 \leq \frac{1}{2}$, we know that

$$D_2 \leq 2\left((\kappa_A + \kappa_B\kappa)D_1 + \frac{2(1 + \gamma)W}{\gamma}\right)(1 + \exp(\kappa_A)).$$

Using the notation

$$\gamma_1 = 2h(\kappa_A + \kappa_B\kappa)(1 + \exp(\kappa_A)) \,,$$

$$\gamma_2 = \frac{(l\kappa_B + 1)2\kappa^2 W}{\gamma} + 4\frac{(1 + \gamma)W}{\gamma}(1 + \exp(\kappa_A))\left(\frac{(l\kappa_B + 1)\kappa^2\kappa_A}{\gamma} + 1\right) \,.$$

We have $\|x_{t+\zeta}\| \leq \gamma_1 D_1 + \gamma_2$.

From the equation of $\gamma_1$ we know that when $h \leq \frac{1}{4(\kappa_A + \kappa_B\kappa)(1 + \exp(\kappa_A))}$ we have $\gamma_1 \leq \frac{1}{2}$. Then we choose $D_1 = 2\gamma_2$, we finally get

$$\|x_{t+\zeta}\| \leq \gamma_1 D_1 + \gamma_2 \leq D_1 \,.$$

Finally, set

$$h_0 = \min\left\{1, \frac{\gamma}{\kappa_A(1 + \exp(\kappa_A))}, \frac{1}{4(\kappa_A + \kappa_B\kappa)(1 + \exp(\kappa_A))}\right\} \,,$$

By the relationship $D_1 = 2\gamma_2$, $D_2 = 2\beta_2$, $D_3 = \kappa_A D_2 + W$, $W_0 = W + hD_3$,

we can verify the induction hypothesis. Moreover, we know that $D_1, D_2, D_3$ are not depend on $h$. Therefore we have proved the claim.

$\square$

The last step is then to bound the action and the approximation errors of states and actions.

**Lemma 2.** *Under Assumption 1 and 2, choosing arbitrary $h$ in the interval $[0, h_0]$ where $h_0$ is a constant only depends on the parameters in the assumption, we have for any $t$ and policy $M_i$, $\|x_t\|, \|y_t\|, \|u_t\|, \|v_t\| \leq D$, $\|\dot{x}_t\| \leq D$, $\|x_t - y_t\|, \|u_t - v_t\| \leq \kappa^2(1 + \kappa)(1 - h\gamma)^{Hm+1}D$. In particular, taking all the $M_t = 0$ and $K = K^*$, we can also obtain the inequality of the optimal solution: $\|x_t^*\|, \|u_t^*\| \leq D$.*

*Proof.* By Lemma 8, we have

$$\|\Psi_{t,i}\| \leq a(lh\kappa_B + 1)\kappa^2(1 - h\gamma)^{i-1} \,.$$

By Lemma 9 we know that for any $h$ in $[0, h_0]$, where

$$h_0 = \min\left\{1, \frac{\gamma}{\kappa_A(1 + \exp(\kappa_A))}, \frac{1}{4(\kappa_A + \kappa_B\kappa)(1 + \exp(\kappa_A))}\right\} \,,$$

we have $\|x_t\| \le D_1$.

By Lemma 7, Lemma 8 and Lemma 9, we have

$$\|y_{t+1}\| = \|h \sum_{i=0}^{2Hm} \Psi_{t,i} \hat{w}_{t-i}\|$$

$$\le h W_0 \sum_{i=0}^{2Hm} a(lh\kappa_B + 1)\kappa^2 (1 - h\gamma)^{i-1}$$

$$\le \frac{a W_0 (lh\kappa_B + 1)\kappa^2}{\gamma} = \tilde{D}_1 .$$

Via the definition of $x_t, y_t$, we have

$$\|x_t - y_t\| \le \kappa^2 (1 - h\gamma)^{Hm+1} \|x_{t-Hm}\| \le \kappa^2 (1 - h\gamma)^{Hm+1} D_1 .$$

For the actions

$$u_t = -Kx_t + h \sum_{i=1}^{Hm} M_t^i \hat{w}_{t-i} ,$$

$$v_t = -Ky_t + h \sum_{i=1}^{Hm} M_t^i \hat{w}_{t-i} ,$$

we can derive the bound

$$\|u_t\| \le \|Kx_t\| + h \sum_{i=1}^{Hm} \|M_t^i \hat{w}_{t-i}\| \le \kappa \|x_t\| + W_0 h \sum_{i=1}^{Hm} a(1 - h\gamma)^{i-1} \le \kappa D_1 + \frac{a W_0}{\gamma} ,$$

$$\|v_t\| \le \|Ky_t\| + h \sum_{i=1}^{Hm} \|M_t^i \hat{w}_{t-i}\| \le \kappa \|y_t\| + W_0 h \sum_{i=1}^{Hm} a(1 - h\gamma)^{i-1} \le \kappa \tilde{D}_1 + \frac{a W_0}{\gamma} ,$$

$$\|u_t - v_t\| \le \|K\| \|x_t - y_t\| \le \kappa^3 (1 - h\gamma)^{Hm+1} D_1 .$$

By Lemma 9, taking $D = \max\{D_1, D_2, \tilde{D}_1, \kappa D_1 + \frac{W_0}{\gamma}, \kappa \tilde{D}_1 + \frac{W_0}{\gamma}\}$, we get the following inequality: $\|x_t\|, \|y_t\|, \|u_t\|, \|v_t\| \le D, \|\dot{x}_t\| \le D$.

We also have

$$\|x_t - y_t\| + \|u_t - v_t\| \le \kappa^2 (1 - h\gamma)^{Hm+1} D_1 + \kappa^3 (1 - h\gamma)^{Hm+1} D_1 \le \kappa^2 (1 + \kappa)(1 - h\gamma)^{Hm+1} D .$$

In particular, the optimal policy can be recognized as taking the DAC policy with all the $M_t$ equal to 0 and the fixed strongly stable policy $K = K^*$. So we also have $\|x_t^*\|, \|u_t^*\| \le D$.

$\square$

Now we have finished the analysis of evolution of the states. It will be helpful to prove the key lemmas in this paper.

## D  Proof of Lemma 3

In this section we will prove the following lemma:

**Lemma 3.** *Under Assumption 2, Algorithm 1 attains the following bound of $R_0$:*

$$R_0 = \sum_{i=0}^{n-1} \int_{ih}^{(i+1)h} (c_t(x_t, u_t) - c_t(x_t^*, u_t^*))dt - h \sum_{i=0}^{n-1} (c_i(x_i, u_i) - c_i(x_i^*, u_i^*)) \le (G+L)D^2 hT .$$

*Proof.* By Assumption 2 and Lemma 2, since we use the unchanged policy $u_t$ in the interval $t \in [ih, (i+1)h]$, we have

$$|c_t(x_t, u_t) - c_{ih}(x_{ih}, u_{ih})| \leq |c_t(x_t, u_t) - c_t(x_{ih}, u_{ih})| + |c_t(x_{ih}, u_{ih}) - c_{ih}(x_{ih}, u_{ih})|$$
$$\leq \max_x \|\nabla_x c_t(x, u)\| \|x_t - x_{ih}\| + L(t - ih)D^2$$
$$\leq GD\| \int_{ih}^t \dot{x}_s ds \| + L(t - ih)D^2$$
$$\leq (G + L)D^2(t - ih).$$

Therefore we have

$$|\sum_{i=0}^{n-1} \int_{ih}^{(i+1)h} c_t(x_t, u_t) dt - h \sum_{i=0}^{n-1} c_i(x_i, u_i)|$$
$$=|\sum_{i=0}^{n-1} \int_{ih}^{(i+1)h} (c_t(x_t, u_t) - c_{ih}(x_{ih}, u_{ih})) dt|$$
$$\leq (G + L)D^2 \sum_{i=0}^{n-1} \int_{ih}^{(i+1)h} (t - ih) dt = \frac{1}{2}(G + L)D^2 nh^2 = \frac{1}{2}(G + L)D^2 hT.$$

A similar bound can easily be established by lemma 2 about the optimal state and policy:

$$|\sum_{i=0}^{n-1} \int_{ih}^{(i+1)h} c_t(x_t^*, u_t^*) dt - \sum_{i=0}^{n-1} c_i(x_i^*, u_i^*)| \leq \frac{1}{2}(G + L)D^2 hT.$$

Taking sum of the two terms we get $R_0 \leq (G + L)D^2 hT$.

$\square$

## E  Proof of Lemma 4

In this section we will prove the following lemma:

**Lemma 4.** *Under Assumption 1 and 2, Algorithm 1 attains the following bound of $R_1$:*

$$R_1 = \sum_{i=0}^{p-1} \left( \sum_{j=im}^{(i+1)m-1} c_i(x_i, u_i) - f_i\left(\tilde{M}_{i-H}, \ldots, \tilde{M}_i\right) \right) \leq nGD^2\kappa^2(1 + \kappa)(1 - h\gamma)^{Hm+1}.$$

*Proof.* Using Lemma 2 and Assumption 2, have the approximation error between ideal cost and actual cost bounded as,

$$|c_t(x_t, u_t) - c_t(y_t, v_t)| \leq |c_t(x_t, u_t) - c_t(y_t, u_t)| + |c_t(y_t, u_t) - c_t(y_t, v_t)|$$
$$\leq GD\|x_t - y_t\| + GD\|u_t - v_t\|$$
$$\leq GD^2\kappa^2(1 + \kappa)(1 - h\gamma)^{Hm+1},$$

where the first inequality is by triangle inequality, the second inequality is by Assumption 2, Lemma 2, and the third inequality is by Lemma 2.

With this, we have

$$R_1 = \sum_{i=0}^{p-1} \left( \sum_{j=im}^{(i+1)m-1} c_i(x_i, u_i) - f_i(\tilde{M}_{i-H}, ..., \tilde{M}_i) \right)$$
$$= \sum_{i=0}^{p-1} \left( \sum_{j=im}^{(i+1)m-1} c_i(x_i, u_i) - \sum_{j=im}^{(i+1)m-1} c_i(y_i, v_i) \right)$$
$$\leq \sum_{i=0}^{p-1} \sum_{j=im}^{(i+1)m-1} GD^2\kappa^2(1 + \kappa)(1 - h\gamma)^{Hm+1} \leq nGD^2\kappa^2(1 + \kappa)(1 - h\gamma)^{Hm+1}.$$

$\square$

# F Proof of Lemma 5

Before we start the proof of Lemma 5, we first present an overview of the online convex optimization (OCO) with memory framework. Consider the setting where, for every $t$, an online player chooses some point $x_t \in \mathcal{K} \subset \mathbb{R}^d$, a loss function $f_t : \mathcal{K}^{H+1} \mapsto \mathbb{R}$ is revealed, and the learner suffers a loss of $f_t(x_{t-H}, \dots, x_t)$. We assume a certain coordinate-wise Lipschitz regularity on $f_t$ of the form such that, for any $j \in \{1, \dots, H\}$, for any $x_1, \dots, x_H, \tilde{x}_j \in \mathcal{K}$

$$|f_t(x_1, \dots, x_j, \dots, x_H) - f_t(x_1, \dots, \tilde{x}_j, \dots, x_H)| \leq L\|x_j - \tilde{x}_j\| .$$

In addition, we define $\tilde{f}_t(x) = f_t(x, \dots, x)$, and we let

$$G_f = \sup_{t \in \{1, \dots, T\}, x \in \mathcal{K}} \left\|\nabla \tilde{f}_t(x)\right\|, \quad D_f = \sup_{x,y \in \mathcal{K}} \|x - y\| .$$

The resulting goal is to minimize the policy regret, which is defined as

$$\text{Regret} = \sum_{t=H}^{T} f_t(x_{t-H}, \dots, x_t) - \min_{x \in \mathcal{K}} \sum_{t=H}^{T} f_t(x, \dots, x) .$$

---

**Algorithm 2** Online Gradient Descent with Memory (OGD-M)

---

Input: Step size $\eta$, functions $\{f_t\}_{t=m}^{T}$.
Initialize $x_0, \dots, x_{H-1} \in \mathcal{K}$ arbitrarily.
**for** $t = H, \dots, T$ **do**
    Play $x_t$, suffer loss $f_t(x_{t-H}, \dots, x_t)$.
    Set $x_{t+1} = \Pi_{\mathcal{K}}\left(x_t - \eta \nabla \tilde{f}_t(x)\right)$.
**end for**

---

To minimize this regret, a commonly used algorithm is the Online Gradient descent. By running the Algorithm 2, we may bound the policy regret by the following lemma:

**Lemma 10.** *Let $\{f_t\}_{t=1}^{T}$ be Lipschitz continuous loss functions with memory such that $\tilde{f}_t$ are convex. Then by runnning algorithm 2 itgenerates a sequence $\{x_t\}_{t=1}^{T}$ such that*

$$\sum_{t=H}^{T} f_t(x_{t-H}, \dots, x_t) - \min_{x \in \mathcal{K}} \sum_{t=H}^{T} f_t(x, \dots, x) \leq \frac{D_f^2}{\eta} + TG_f^2\eta + LH^2\eta G_f T .$$

*Furthermore, setting $\eta = \dfrac{D_f}{\sqrt{G_f(G_f + LH^2)T}}$ implies that*

$$\text{PolicyRegret} \leq 2D_f\sqrt{G_f(G_f + LH^2)T} .$$

*Proof.* By the standard OGD analysis [18], we know that

$$\sum_{t=H}^{T} \tilde{f}_t(x_t) - \min_{x \in \mathcal{K}} \sum_{t=H}^{T} \tilde{f}_t(x) \leq \frac{D_f^2}{\eta} + TG^2\eta.$$

In addition, we know by the Lipschitz property, for any $t \geq H$, we have

$$|f_t(x_{t-H}, \dots, x_t) - f_t(x_t, \dots, x_t)| \leq L\sum_{j=1}^{H}\|x_t - x_{t-j}\| \leq L\sum_{j=1}^{H}\sum_{l=1}^{j}\|x_{t-l+1} - x_{t-l}\|$$

$$\leq L\sum_{j=1}^{H}\sum_{l=1}^{j}\eta\left\|\nabla \tilde{f}_{t-l}(x_{t-l})\right\| \leq LH^2\eta G,$$

and so we have that

$$\left|\sum_{t=H}^{T} f_t(x_{t-H}, \dots, x_t) - \sum_{t=H}^{T} f_t(x_t, \dots, x_t)\right| \leq TLH^2\eta G.$$

It follows that
$$\sum_{t=H}^{T} f_t\left(x_{t-H}, \ldots, x_t\right) - \min_{x \in \mathcal{K}} \sum_{t=H}^{T} f_t(x, \ldots, x) \le \frac{D_f^2}{\eta} + TG_f^2 \eta + LH^2 \eta G_f T\,.$$

$\square$

In this setup, the first term corresponds to the DAC policy we make, and the second term is used to approximate the optimal strongly stable linear policy. It is worth noting that the cost of OCO with memory depends on the update frequency $H$. Therefore, we propose a two-level online controller. The higher-level controller updates the policy with accumulated feedback at a low frequency to reduce the regret, whereas a lower-level controller provides high-frequency updates of the DAC policy to reduce the discretization error. In the following part, we define the update distance of the DAC policy as $l = Hm$, where $m$ is the ratio of frequency between the DAC policy update and OCO memory policy update. Formally, we update the value of $M_t$ once every $m$ transitions, where $g_t$ represents a loss function.
$$M_{t+1} = \begin{cases} \Pi_{\mathcal{M}}\left(M_t - \eta \nabla g_t(M)\right) & \text{if } t\%m == 0 \\ M_t & \text{otherwise}\,. \end{cases}$$

From now on, we denote $\tilde{M}_t = M_{tm}$ for the convenience to remove the duplicate elements. By the definition of ideal cost, we know that it is a well-defined definition.

By Lemma 7 we know that
$$y_{t+1} = h \sum_{i=0}^{2Hm} \Psi_{t,i} \hat{w}_{t-i},$$
$$v_t = -K y_t + h \sum_{j=1}^{Hm} M_t^j \hat{w}_{t-j}\,,$$

where
$$\Psi_{t,i} = Q_h^i \mathbf{1}_{i \le l} + h \sum_{j=0}^{l} Q_h^j B M_{t-j}^{i-j} \mathbf{1}_{i-j \in [1,l]}\,.$$

So we know that $y_t$ and $y_t$ are linear combination of $M_t$, therefore
$$f_i\left(\tilde{M}_{i-H}, \ldots, \tilde{M}_i\right) = \sum_{t=im}^{(i+1)m-1} c_t\left(y_t\left(\tilde{M}_{i-H}, \ldots, \tilde{M}_i\right), v_t\left(\tilde{M}_{i-H}, \ldots, \tilde{M}_i\right)\right)\,.$$

is convex in $M_t$. So we can use the OCO with memory structure to solve this problem.

By Lemma 9 we know that $y_t$ and $v_t$ are bounded by $D$. Then we need to calculate the diameter, Lipchitz constant, and gradient bound of this function $f_i$. In the following, we choose the DAC policy parameter $l = Hm$.

**Lemma 11.** *(Bounding the diameter) We have*
$$D_f = \sup_{M_i, M_j \in \mathcal{M}} \|M_i - M_j\| \le \frac{2a}{h\gamma}$$

.

*Proof.* By the definition of $\mathcal{M}$, taking $l = Hm$ we know that

$$\sup_{M_i, M_j \in \mathcal{M}} \|M_i - M_j\| \le \sum_{k=1}^{Hm} \|M_i^k - M_j^k\|$$
$$\le \sum_{k=1}^{Hm} 2a(1 - h\gamma)^{k-1}$$
$$\le \frac{2a}{h\gamma}\,.$$

$\square$

**Lemma 12.** *(Bounding the Lipschitz Constant) Consider two policy sequences* $\left\{ \tilde{M}_{i-H} \ldots \tilde{M}_{i-k} \ldots \tilde{M}_i \right\}$ *and* $\left\{ \tilde{M}_{i-H} \ldots \hat{M}_{i-k} \ldots \tilde{M}_i \right\}$ *which differ in exactly one policy played at a time step* $t - k$ *for* $k \in \{0, \ldots, H\}$. *Then we have that*

$$\left| f_i\left( \tilde{M}_{i-H} \ldots \tilde{M}_{i-k} \ldots \tilde{M}_i \right) - f_i\left( \tilde{M}_{i-H} \ldots \hat{M}_{i-k} \ldots \tilde{M}_i \right) \right| \leq C^2 \kappa^3 \kappa_B W_0 \sum_{j=0}^{Hm} \| \tilde{M}_{i-k}^j - \hat{M}_{i-k}^j \|,$$

*where* $C$ *is a constant.*

*Proof.* By the definition we have

$$\|y_t - \tilde{y}_t\| = \| h \sum_{i=0}^{2Hm} h \sum_{j=0}^{Hm} Q_h^j B(M_{t-j}^{i-j} - \tilde{M}_{t-j}^{i-j}) \mathbf{1}_{i-j \in [1, Hm]} \hat{w}_{t-i} \|$$

$$\leq h^2 \kappa^2 \kappa_B W_0 \sum_{i=0}^{2Hm} \sum_{j=0}^{Hm} \| M_{t-j}^{i-j} - \tilde{M}_{t-j}^{i-j} \| \mathbf{1}_{i-j \in [1, Hm]}$$

$$\leq h^2 \kappa^2 \kappa_B W_0 m \sum_{j=0}^{Hm} \| \tilde{M}_{i-k}^j - \hat{M}_{i-k}^j \|$$

$$= hC\kappa^2 \kappa_B W_0 \sum_{j=0}^{Hm} \| \tilde{M}_{i-k}^j - \hat{M}_{i-k}^j \|.$$

Where the first inequality is by $\|Q_h^j\| \leq \kappa^2 (1 - h\gamma)^{j-1} \leq \kappa^2$ and lemma 9 of bounded estimation disturbance, the second inequality is by the fact that $M_{i-k}$ have taken $m$ times, the last equality is by $m = \frac{C}{h}$. Furthermore, we have that

$$\|v_t - \tilde{v}_t\| = \| - K(y_t - \tilde{y}_t) \| \leq hC\kappa^3 \kappa_B W_0 \sum_{j=0}^{Hm} \left\| \tilde{M}_{i-k}^j - \hat{M}_{i-k}^j \right\|.$$

Therefore using Assumption 2, Lemma 9 and Lemma 2 we immediately get that

$$\left| f_i\left( \tilde{M}_{i-H} \ldots \tilde{M}_{i-k} \ldots \tilde{M}_i \right) - f_i\left( \tilde{M}_{i-H} \ldots \hat{M}_{i-k} \ldots \tilde{M}_i \right) \right| \leq C^2 \kappa^3 \kappa_B W_0 \sum_{j=0}^{Hm} \| \tilde{M}_{i-k}^j - \hat{M}_{i-k}^j \|.$$

$\square$

**Lemma 13.** *(Bounding the Gradient) We have the following bound for the gradient:*

$$\|\nabla_M f_t(M \ldots M)\|_F \leq \frac{GDC\kappa^2(\kappa + 1)W_0 \kappa_B}{\gamma}$$

*Proof.* Since $M$ is a matrix, the $\ell_2$ norm of the gradient $\nabla_M f_t$ corresponds to the Frobenius norm of the $\nabla_M f_t$ matrix. So it will be sufficient to derive an absolute value bound on $\nabla_{M_{p,q}^{[r]}} f_t(M, \ldots, M)$ for all $r, p, q$. To this end, we consider the following calculation. Using lemma 9 we get that $y_t(M \ldots M), v_t(M \ldots M) \leq D$. Therefore, using Assumption 2 we have that

$$\left| \nabla_{M_{p,q}^{[r]}} c_t(M \ldots M) \right| \leq GD \left( \left\| \frac{\partial y_t(M)}{\partial M_{p,q}^{[r]}} + \frac{\partial v_t(M \ldots M)}{\partial M_{p,q}^{[r]}} \right\| \right).$$

We now bound the quantities on the right-hand side:

$$\left\|\frac{\delta y_t(M\ldots M)}{\delta M_{p,q}^{[r]}}\right\| = \left\|h\sum_{i=0}^{2Hm}h\sum_{j=1}^{Hm}\left[\frac{\partial Q_h^j BM^{[i-j]}}{\partial M_{p,q}^{[r]}}\right]\hat{w}_{t-i}\mathbf{1}_{i-j\in[1,H]}\right\|$$

$$\leq h^2\sum_{i=r}^{r+Hm}\left\|\left[\frac{\partial Q_h^{i-r}BM^{[r]}}{\partial M_{p,q}^{[r]}}\right]w_{t-i}\right\|$$

$$\leq h^2\kappa^2 W_0\kappa_B\frac{1}{h\gamma} = \frac{h\kappa^2 W_0\kappa_B}{\gamma}\,.$$

Similarly,

$$\left\|\frac{\partial v_t(M\ldots M)}{\partial M_{p,q}^{[r]}}\right\| \leq \kappa\left\|\frac{\delta y_t(M\ldots M)}{\delta M_{p,q}^{[r]}}\right\| \leq \kappa\frac{h\kappa^2 W_0\kappa_B}{\gamma} \leq \frac{h\kappa^3 W_0\kappa_B}{\gamma}\,.$$

Combining the above inequalities with

$$f_i\left(\tilde{M}_{i-H},\ldots,\tilde{M}_i\right) = \sum_{t=im}^{(i+1)m-1}c_t\left(y_t\left(\tilde{M}_{i-H},\ldots,\tilde{M}_i\right),v_t\left(\tilde{M}_{i-H},\ldots,\tilde{M}_i\right)\right)\,.$$

gives the bound that

$$\|\nabla_M f_t(M\ldots M)\|_F \leq \frac{GDC\kappa^2(\kappa+1)W_0\kappa_B}{\gamma}\,.$$

$\square$

Finally we prove Lemma 5:

**Lemma 5.** *Under Assumption 1 and 2, choosing $m = \frac{C}{h}$ and $\eta = \Theta(\frac{m}{Th})$, Algorithm 1 attains the following bound of $R_2$:*

$$R_2 = \sum_{i=0}^{p-1}f_i(\tilde{M}_{i-H},\ldots,\tilde{M}_i) - \min_{M\in\mathcal{M}}\sum_{i=0}^{p-1}f_i(M,\ldots,M)$$

$$\leq \frac{4a}{\gamma}\sqrt{\frac{GDC^2\kappa^2(\kappa+1)W_0\kappa_B}{\gamma}\left(\frac{GDC\kappa^2(\kappa+1)W_0\kappa_B}{\gamma} + C^2\kappa^3\kappa_B W_0 H^2\right)\frac{n}{h}}\,.$$

*Proof.* By Lemma 10 we have

$$R_2 \leq 2D_f\sqrt{G_f\left(G_f + LH^2\right)p}$$

By Lemma 11, Lemma 12, and Lemma 13 we have

$$R_2 \leq 2D_f\sqrt{G_f\left(G_f + LH^2\right)p}$$

$$\leq 2\frac{2a}{h\gamma}\sqrt{\frac{GDC\kappa^2(\kappa+1)W_0\kappa_B}{\gamma}\left(\frac{GDC\kappa^2(\kappa+1)W_0\kappa_B}{\gamma} + C^2\kappa^3\kappa_B W_0 H^2\right)\frac{n}{m}}$$

$$\leq \frac{4a}{\gamma}\sqrt{\frac{GDC^2\kappa^2(\kappa+1)W_0\kappa_B}{\gamma}\left(\frac{GDC\kappa^2(\kappa+1)W_0\kappa_B}{\gamma} + C^2\kappa^3\kappa_B W_0 H^2\right)\frac{n}{h}}\,.$$

$\square$

# G  Proof of Lemma 6

In this section, we will prove the approximation value of DAC policy and optimal policy is sufficiently small. First, we introduce the following:

**Lemma 14.** *For any two $(\kappa, \gamma)$-strongly stable matrices $K^*, K$, there exists $M = \left( M^1, \ldots, M^{Hm} \right)$ where*

$$M^i = (K - K^*) \left( I + h(A - BK^*) \right)^{i-1} ,$$

*such that*

$$c_t(x_t(M), u_t(M)) - c_t(x_t^*, u_t^*) \leq GDW_0 \kappa^3 a(lh\kappa_B + 1)(1 - h\gamma)^{Hm} .$$

*Proof.* Denote $Q_h(K) = I + h(A - BK)$, $Q_h(K^*) = I + h(A - BK^*)$. By Lemma 7 we have

$$x_{t+1}^* = h \sum_{i=0}^{t} Q_h^i(K^*) \hat{w}_{t-i} .$$

Consider the following calculation for $i \leq Hm$ and $M^i = (K - K^*) \left( I + h(A - BK^*) \right)^{i-1}$:

$$
\begin{aligned}
\Psi_{t,i} \left( M, \ldots, M \right) &= Q_h^i(K) + h \sum_{j=1}^{i} Q_h^{i-j}(K) BM^j \\
&= Q_h^i(K) + h \sum_{j=1}^{i} Q_h^{i-j}(K) B \left( K - K^* \right) Q_h^{j-1}(K^*) \\
&= Q_h^i(K) + \sum_{j=1}^{i} Q_h^{i-j}(K)(Q_h(K^*) - Q_h(K)) Q_h^{j-1}(K^*) \\
&= Q_h^i(K^*) ,
\end{aligned}
$$

where the final equality follows as the sum telescopes. Therefore, we have that

$$x_{t+1}(M) = h \sum_{i=0}^{Hm} Q_h^i(K^*) \hat{w}_{t-i} + h \sum_{i=Hm+1}^{t} \Psi_{t,i} \hat{w}_{t-i} .$$

Then we obtain that

$$\left\| x_{t+1}(M) - x_{t+1}^* \right\| \leq hW_0 \sum_{i=Hm+1}^{t} \left( \left\| \Psi_{t,i} \left( M_* \right) \right\| + \left\| Q_h^i(K^*) \right\| \right) .$$

Using Definition 1 and Lemma 7 we finally get

$$
\begin{aligned}
\left\| x_{t+1}(M) - x_{t+1}^* \right\| &\leq hW_0 \Big( \sum_{i=Hm+1}^{t} \left( (lh\kappa_B + 1) a\kappa^2 (1 - h\gamma)^{i-1} \right) + \kappa^2 (1 - h\gamma)^i \Big) \\
&\leq W_0 (lh\kappa_B + 2) a\kappa^2 (1 - h\gamma)^{Hm} .
\end{aligned}
$$

We also have

$$\|u_t^* - u_t\,(M)\| = \left\| -K^* x_t^* + K x_t\,(M) - h\sum_{i=0}^{Hm} M^i \hat{w}_{t-i} \right\|$$

$$= \|(K - K^*)x_t^* + K(x_t(M) - x_t^*) - h\sum_{i=0}^{Hm} M^i \hat{w}_{t-i}\|$$

$$= \|(K - K^*)h\sum_{i=0}^{t-1} Q_h^i(K^*)\hat{w}_{t-i} + K(x_t(M) - x_t^*) - h\sum_{i=0}^{Hm} M^i \hat{w}_{t-i}\|$$

$$= \|K(x_t(M) - x_t^*) - h\sum_{i=Hm+1}^{t-1} (K - K^*)Q_h^{i-1}(K^*)\hat{w}_{t-i}\|$$

$$= \|Kh\sum_{i=Hm+1}^{t-1} (\Psi_{t,i} - Q_h^{i-1}(K^*))\hat{w}_{t-i} - h\sum_{i=Hm+1}^{t-1} (K - K^*)Q_h^{i-1}(K^*)\hat{w}_{t-i}\|$$

$$= \left\| h\sum_{i=Hm+1}^{t-1} K^* \left(Q_h^{i-1}(K^*) + \Psi_{t,i}\right) \hat{w}_{t-i} \right\|$$

$$\leq W_0\kappa((1 - h\gamma)^{Hm} + a(lh\kappa_B + 1)\kappa^2(1 - h\gamma)^{Hm})$$

$$= W_0\kappa(a(lh\kappa_B + 1)\kappa^2 + 1)(1 - h\gamma)^{Hm},$$

where the inequality is by Definition 1 and Lemma 8.

Finally, we have

$$|c_t\,(x_t(M), u_t(M)) - c_t\,(x_t^*, u_t^*)|$$
$$\leq |c_t\,(x_t(M), u_t(M)) - c_t\,(x_t^*, u_t(M))| + |c_t\,(x_t^*, u_t(M)) - c_t\,(x_t^*, u_t^*)|$$
$$\leq GD|x_t(M) - x_t^*| + GD|u_t(M) - u_t^*|$$
$$\leq GDW_0\kappa^3 a(lh\kappa_B + 1)(1 - h\gamma)^{Hm},$$

where the second inequality is by Assumption 2. $\qquad\square$

Then we can prove our main lemma:

**Lemma 6.** *Under Assumption 1 and 2, Algorithm 1 attains the following bound of $R_3$:*

$$R_3 = \min_{M\in\mathcal{M}} \sum_{i=0}^{p-1} f_i(M, ..., M) - \sum_{i=0}^{p-1}\sum_{j=im}^{(i+1)m-1} c_i(x_i^*, u_i^*) \leq 3n(1 - h\gamma)^{Hm}GDW_0\kappa^3 a(lh\kappa_B + 1).$$

*Proof.* By choosing

$$M^i = (K - K^*)\left(I + h(A - BK^*)\right)^{i-1}.$$

We know that

$$\|M^i\| = \|(K - K^*)\left(I + h(A - BK^*)\right)^{i-1}\| \leq 2\kappa^3(1 - \gamma)^{i-1}.$$

Therefore choose $a = 2\kappa^3$ we have $M = \{M^i\}$ in the DAC policy update class $\mathcal{M}$.

Then we have the analysis of the regret:

$$R_3 = \min_{M\in\mathcal{M}} \sum_{i=0}^{p-1} f_i(M, ..., M) - \sum_{i=0}^{p-1}\sum_{j=im}^{(i+1)m-1} c_i(x_i^*, u_i^*)$$

$$\leq \min_{M\in\mathcal{M}} \sum_{i=0}^{p-1}\sum_{j=im}^{(i+1)m-1} c_i(x_i(M), u_i(M)) - \sum_{i=0}^{p-1}\sum_{j=im}^{(i+1)m-1} c_i(x_i^*, u_i^*) + n\kappa^2(1 + \kappa)(1 - h\gamma)^{Hm+1}D$$

$$\leq 3n(1 - h\gamma)^{Hm}GDW_0\kappa^3 a(lh\kappa_B + 1),$$

where the first inequality is by Lemma 2 and the second inequality is by Lemma 14.

$\qquad\square$

