# OpenReview forum: "Online Control with Adversarial Disturbance for Continuous-time Linear Systems"
_NeurIPS.cc/2024/Conference — NeurIPS 2024 poster_

### Official Review · Reviewer_3oRE · 2024-06-25

**Soundness:** 2
**Presentation:** 3
**Contribution:** 2
**Rating:** 4
**Confidence:** 4

**Summary:**

This paper considers the problem of online non-stochastic control within continuous-time linear dynamical systems. To this end, the authors proposed a two-level online algorithm, where the higher level deals with a discretized reduced problem to minimize the performance measure of regret, while the lower level handles the discretization error. This work achieves an $O(\sqrt{T} \log T)$ regret, which is the same as that in the discretized system. Finally, empirical evaluations validate the effectiveness of the proposed method.

**Strengths:**

This work investigates the meaningful problem of controlling a continuous-time linear dynamical system. This work provides an $O(\sqrt{T} \log T)$ regret, which is in the same order compared with the result in discretized systems. Furthermore, the authors have conducted sufficient experiments to study the effects of frame stack and frame skip, and show the effectiveness of the proposed method.

**Weaknesses:**

Although this work has done a nice job in handling continuous-time systems in the online non-stochastic control setup, my major concern is that the technical novelty seems insufficient.

Specifically, as the authors have said, two technical challenges exist to achieve the main Theorem 1. The first one is to control the magnitude of the states, and the second one is to handle the curse of dimensionality caused by discretization. As for the first technical challenge, the idea of "employ Gronwall’s inequality to bound the first and second-order derivatives in the neighborhood of the current state" (Line 207-209) seems to be pretty natural because of the continuous nature of the system. And the second technical challenge seems to be leveraging the *lazy-update* paradigm to facilitate some kind of parameter tuning for the desired regret bound. Both technical challenges seem to be a little minor and thus insufficient from my point of view. Could the authors provide more detailed explanations on the technical challenges of dealing with continuous-time systems compared with the analysis in a discretized one?

**Questions:**

In this part, I list some other questions or suggestions:
1. In Line 90, the authors said "However, it is unclear why domain randomization works in theory". However, this work does not illuminate the role of domain randomization in theory either because the DR only appears in the experiments part. I suggest that the authors could revise the sentence to avoid unnecessary misunderstanding of this work's contributions.
2. Section 3.2, in my opinion, could be put in the appendix because the relationship with robust control should be explained as a part of the related work, but not necessarily to be mentioned when introducing the problem setup of *this* work.
3. Assumption 1, compared with the standard non-stochastic disturbances in discretized systems, uses an extra condition of $\|\dot{w}_t\| \le W$. I guess that this additional condition is used to solve the first technical challenge (employ Gronwall’s inequality to bound the first and second-order derivatives in the neighborhood of the current state)? Am I right? Besides, the comparison of Assumption 1 with the previous one should be explicitly explained in the next version.
4. Definition 1 is a little different from the standard strong stability condition in discretized systems. In discretized systems, a linear controller is strongly stable if $A - BK = P L P^{-1}$. However, in this work, the strong stability holds for $I + h(A-BK)$. By setting $h=1$ as a special case, this assumption is still different with an extra identity matrix $I$. Why are they different? And the comparison with the standard discretized strong stability assumption should be explained in the next version.
5. In Section 4, some citations of the pioneering work [Agarwal et al., ICML 2019] are missing. For example,  Definition 2,3,4 should cite [Agarwal et al., ICML 2019]. I understand that there exist minor differences in the definitions in this work compared with those in [Agarwal et al., ICML 2019] for discretized systems. However, correct citations should still be provided to avoid unnecessary misunderstandings of this work's contributions.
6. In the equation after Line 172, the notation of '$g_t$' is not defined. I noticed that $g_t$ is first defined in Algorithm 1 on the next page.
7. In Figure 2, the font sizes of 'Replay' and 'Buffer' seem to be different?

**Limitations:**

The authors have adequately addressed the limitations and, if applicable, potential negative societal impact of their work.

---

> ### Author Rebuttal · Authors · 2024-08-07
>
> We extend our gratitude to the reviewer for the comments and suggestions.  Below, we address the primary concern that has been raised. The minor issues and typographical errors have been corrected in our manuscript.
>
>
> >Q1: Could the authors provide more detailed explanations on the technical challenges of dealing with continuous-time systems compared with the analysis in a discretized one?
>
> **A1:** We appreciate the reviewer for raising this question and we will provide the main technical challenge here. Both of these challenges arise when **extending from discrete systems to continuous systems** and we can not directly apply the method in [1] which is only suitable for discrete system.
>
> For the first technical challenge of unbounded states in continuous system, as in discrete system it is straightforward to demonstrate that the state sequence by applying the dynamics inequality $\|x_{t+1}\| \le a\|x_t\| + b$ and the induction method presented in [1]. In continuous system, one naive approach is to use the Taylor expansion of each state to derive the recurrence formula of the state. However, this argument **requires the prerequisite knowledge that the states within this neighborhood are bounded by the dynamics**, leading to circular reasoning. We use Gronwall's inequality to overcome this challenge.
>
> For the second challenge, initially we set out to follow the approach from [1], where the OCO parameters are updated at each step. Our regret is **primarily composed of three components**:  the error caused by discretization $R_1$, the regret of OCO with memory $R_2$ and the difference between the actual cost and the approximate cost $R_3$. The discretization error $R_1$ is $O(hT)$, therefore if we achieve $O(\sqrt{T})$ regret, we must choose $h$ no more than $O(\frac{1}{\sqrt{T}})$.
>
> If we update the OCO with memory parameter at each timestep follow the method in [1], we will incur the regret of OCO with memory $R_2 = O(H^{2.5}\sqrt{T})$. The difference between the actual cost and the approximate cost $R_3 = O(T(1-h\gamma)^{H})$. To achieve sublinear regret for the third term, we must choose $H = O(\frac{\log T}{h\gamma})$, but since $h$ is **no more than** $O(\frac{1}{\sqrt{T}})$, $H$ will be **larger than** $\Theta(\sqrt{T})$, therefore the second term $R_2$ will definitely exceed $O(\sqrt{T})$.
>
> Therefore, we adjusted the frequency of updating the OCO parameters by introducing a two level algorithm approach, **update the parameters once in every $m$ steps.** This will incur the third term $R_3 = O(T(1-h\gamma)^{Hm})$ but keep the OCO with memory regret $R_2 = O(H^{2.5}\sqrt{T})$, so we can choose $H = O(\frac{\log T}{\gamma})$ and $m=O(\frac{1}{h})$. Then the term of $R_2$ is $O(\sqrt{T}\log T)$ and we achieve the same regret compare with the discrete system.
>
> >Q2: Definition 1 is a little different from the standard strong stability condition in discretized systems.
>
> **A2:** We understand the reviewers' concern regarding the difference in definitions for discrete systems. We will provide further explanation here. For discrete systems, we have the following transition equation:
> $$x_{t+1} = Ax_t+Bu_t + w_t.$$
> For continuous systems, the transition equation is:
> $$\dot{x}_{t} = \tilde{A}x_t+\tilde{B}u_t + \tilde{w}_t.$$
>
> If we consider a relatively small time interval $h$, we can approximate it as follows:
> $$x_{t+h} - x_t = \int_{0}^h \dot{x}_{t+s} ds \approx h( \tilde{A}x_t+\tilde{B}u_t + \tilde{w}_t).$$
>
> Therefore, we have
> $$x_{t+h} \approx (I+h \tilde{A})x_t+h\tilde{B}u_t + h\tilde{w}_t.$$
> Thus, we can express the dynamics of the two systems as follows:
> $$A \approx I+h \tilde{A}, B \approx h\tilde{B}.$$
> As the definition of strongly stable in discrete system is $A-BK = PLP^{-1}$, so we extend the definition to continuous system as $I+h\tilde{A}-h\tilde{B}K = PL_hP^{-1}$, that is $I+h(\tilde{A}-\tilde{B}K) = PL_hP^{-1}$.
>
> >Q3: Assumption 1, compared with the standard non-stochastic disturbances in discretized systems, uses an extra condition of the noise.
>
> **A3:** We understand the reviewers' concern regarding the difference in assumptions for discrete systems. We will provide further explanation here. Similar to the analysis in A2, using the The Taylor formula with Lagrange remainder, we get
> $$
> x_{t+h} = x_t + h \dot{x}_t + \frac{h^2}{2} \ddot{x}_m
> $$
> where $m$ is a number between $[t,t+h]$.
>
> Using the notation and analysis in A2, we get the following relation of discrete and continuous system:
>
> $$
> w_{t} = h \tilde{w}_t + \frac{h^2}{2} \dot{\tilde{w}}_m
> $$
>
> Therefore, if we have the assumption of $\|w_t\|$ is bounded in discrete system, we naturally extend the assumption to $\|\tilde{w}_t\|$ and $\|\dot{\tilde{w}}_t\|$ is bounded in continuous system. We will add some discussion of this assumption in our next version.
>
> >Q4: The arrangement of Section 3.2, missing citations in Section 4, missing definition of $g_t$.
>
> **A4:** We sincerely appreciate the reviewers pointing out these errors, and we will correct them in the next version of the paper.
>
> We thank the reviewer once again for the valuable and helpful suggestions. We would love to provide further clarifications if the reviewer has any additional questions.
>
> **References**
>
> [1] Agarwal, Naman, et al. "Online control with adversarial disturbances." International Conference on Machine Learning. PMLR, 2019.

---

> > ### Comment · Reviewer_3oRE · 2024-08-07
> > **Thanks for the feedback**
> >
> > Thanks for the detailed feedback and explanations. I have read the response carefully, where A2 and A3 have solved my concerns raised in the original review. I believe these assumptions are not stronger than those in the discretized system. I also agree with the technical challenges mentioned in A1.
> >
> > However, in my opinion, the technical contributions are still a little weak. I agree with the first challenge that bounding the magnitude of states requires novel components in continuous systems. However, the second one that the authors have mentioned still seems to be a pretty natural operation, which uses lazy updates to achieve delicated parameter tuning. I do not see much challenge about this point. Besides, I think that the comparisons with previous assumptions in the discretized systems (A2 and A3) are necessary to help readers better understand whether the assumptions here are strong or not. If there are not enough spaces in the main paper, I suggest that the authors could put them in the appendix. Based on this, I am more inclined to believe another round of polishment would be better. Therefore I keep my current score.
> >
> > At last, thank the authors again for the detailed responses.

---

> > > ### Author Response · Authors · 2024-08-08
> > > **Response to further feedback**
> > >
> > > We sincerely thank the reviewer for the rapid and detailed feedback, which will greatly assist us in enhancing the quality of our work.
> > >
> > > We appreciate the reviewer's acknowledgment that our responses to A2 and A3 have addressed your concerns and affirmed that our assumptions are not stronger than those in discretized systems. Although the solution to the second challenge is algorithmically simple, identifying the solution is nontrivial. The key is to see that the learning frequency for online control and the feedback frequency for reducing discretization errors are mismatched. This mismatch is unique to the continuous time system, and up to our knowledge, identified the first time. Overall, our main contribution is that we **provide the first non-asymptotic results for controlling continuous-time linear systems with non-stochastic noise**.
> > >
> > > Regarding your suggestion to include comparisons with previous assumptions in discretized systems (A2 and A3) to aid reader comprehension, we will include these comparisons in the appendix.
> > >
> > > Thank you again for your constructive feedback.

---

### Official Review · Reviewer_H9Pv · 2024-07-12

**Soundness:** 2
**Presentation:** 2
**Contribution:** 3
**Rating:** 6
**Confidence:** 4

**Summary:**

This paper presents an algorithm for the non-stochastic control problem for continuous-time systems. In particular, the proposed algorithm discretizes the continuous-time system and balances the sampling time for discretization and update frequence of the online learning algorithm, in order to obtain a sublinear regret guarantee. The theoretical results are validated in numerical experiments.

**Strengths:**

The paper is well written and studies an interesting research question. In order to obtain their results, the authors make use of novel proof techniques and design ideas for the proposed algorithm.

**Weaknesses:**

In my opinion, the main weakness of this work is that the presentation of the results is often unclear and/or confusing. In particular,
- the way the authors use to denote time steps is very confusing and often $t+1$ and $t+h$ seem to be used interchangeably (compare, e.g., line 8 'Use the action...' in Algorithm 1). Similarly $t$ and $i$ seem to be used interchangeably in Lemma 12. As this paper works with both, continuous and discrete-time, I believe it is highly important to use a notation that is less ambiguous.
- the matrices $M$ of the DAC policy are sometimes used with square brackets (for either the sub- or superscript) and sometimes without apparently without meaning different matrices
- $n$ and $m$ are most frequently used to denote the state and input dimension of the dynamial system, I suggest using different constants in Algorithm 1
- There are many typos in the proofs in the appendix, for example there should not be an 'a' in (2), the constant $a$ is generally not defined (at some point you introduce the condition $|M^i_t | \leq a(1-h\gamma)^{i-1}$, but this does not match the DAC policy class in Algorithm 1?),  I believe the definition of $\gamma_1$ after line 555 is wrong, above line 582, you require $n=\lfloor\frac{T}{h}\rfloor$ contrary to your definition in Algorithm 1 (which, however, you require earlier on in the proof; similar arguments hold for $p$ after line 592), and after line 622, the first sum should start at $i=1$ by definition of $u_t$, whereas it should be $Q_h^i$ instead of $Q_h^{i-1}$ when replacing $x_t(M)-x_t^*$. More generally, I believe that the results are qualitatively correct, but the constants (which are omitted in the $\mathcal O$-notation) are frequently wrong so that I was not able to verify all the presented results

**Questions:**

I am not sure whether I fully understand why updating the DAC policy at each time step (compared to every $m$-th time step as you propose) does not work. Intuitively, I would assume that updating the DAC policy more frequently yields better performance, and therefore also a lower regret bound. After Lemma 10 (page 24) the authors argue 'It is worth noting that the cost of OCO with memory depends on the the update frequency $H$.' However, $H$ is not immediately connected to the update frequency, but rather denotes the size of the memory in the OCO with memory framework described above Lemma 10. Could you please clarify why updating the DAC policy less often leads to improved performance (i.e., lower regret)?

**Limitations:**

From the proof (e.g., after line 549), you seem to assume $x_0=0$. While this assumption is very commonly adopted in the related literature, it should be stated somewhere in the main body of the text explicitly. Furthermore, I am not sure whether this assumption is restrictive: It requires the system to be initialized at i) a steady state of the system and ii) the steady state of the system, that the benchmark policy $K^*$ aims to stabilize.

---

> ### Author Rebuttal · Authors · 2024-08-07
>
> We thank the reviewer for the comments and constructive suggestions. In the following, we focus on explaining why updating policy at each step fail. We also appreciate the comments on notations and will incoporate them in the updated version.
>
>
>
>
> >Q1: Why updating the DAC policy at each time step does not work.
>
> **A1:** We appreciate the reviewer for raising this valuable question. Intuitively, as the disturbance $\omega_t$ is nonstochastic, updating the policy at each discrete step would make the policy unnecessarily sensitive to $\omega_t$ and can incur huge future regret for some worst case disturbance. An alternative would be to use a very small step size $\eta = \Theta(h)$ but large memory size $H = \Theta(1/h)$. However, such an algorithm would not be practical due to the huge memory requirement.
>
>
> We provide more formal explanations below. Initially, we set out to follow the approach from [1], where the OCO parameters are updated at each step. Our regret is **primarily composed of three components**:  the error caused by discretization $R_1$, the regret of OCO with memory $R_2$ and the difference between the actual cost and the approximate cost $R_3$. The discretization error $R_1$ is $O(hT)$, therefore if we achieve $O(\sqrt{T})$ regret, we must choose $h$ no more than $O(\frac{1}{\sqrt{T}})$.
>
> If we update the OCO with memory parameter at each timestep follow the method in [1], we will incur the regret of OCO with memory $R_2 = O(H^{2.5}\sqrt{T})$. The difference between the actual cost and the approximate cost $R_3 = O(T(1-h\gamma)^{H})$. To achieve sublinear regret for the third term, we must choose $H = O(\frac{\log T}{h\gamma})$, but since $h$ is **no more than** $O(\frac{1}{\sqrt{T}})$, $H$ will be **larger than** $\Theta(\sqrt{T})$, therefore the second term $R_2$ will definitely exceed $O(\sqrt{T})$.
>
> Therefore, we adjusted the frequency of updating the OCO parameters by introducing a new parameter $m$, **update the parameters once in every $m$ steps.** This will incur the third term $R_3 = O(T(1-h\gamma)^{Hm})$ but keep the OCO with memory regret $R_2 = O(H^{2.5}\sqrt{T})$, so we can choose $H = O(\frac{\log T}{\gamma})$ and $m=O(\frac{1}{h})$. Then the term of $R_2$ is $O(\sqrt{T}\log T)$ and we achieve the same regret compare with the discrete system.
>
> >Q2: The way the authors use to denote time steps is very confusing and $t+1$ and $t+h$ often seem to be used interchangeably.
>
> **A2:** In lines 155-157 of the paper, we explained that we simplified the subscripts containing "h" to those without "h." However, when recording time, we used the time with "h." In the next version, we will further unify the notation by also writing the time without "h."
>
> >Q3: The matrices of the DAC policy are sometimes used with square brackets and sometimes without. $m$ and $n$ are most frequently used to denote the state and input dimension of the dynamial system, I suggest using different constants in Algorithm 1.
>
> **A3:** In line 161, we explain that the subscripts of the DAC matrix represent the parameter in step $t$, while the superscripts denote each component. Since our paper does not focus on the state and input dimension of the dynamical system and the algorithm is dimension-free, we forgot this issue. We will use more appropriate notation in future versions. We appreciate the reviewer's reminder.
>
>
> >Q4: There are many typos in the proofs in the appendix. Some of the notations in the paper need to be corrected.
>
> **A4:** We greatly appreciate the reviewer for carefully reading our appendix and providing many valuable suggestions for correcting the notation in the paper, and we will carefully check and correct these issues in the next version of the paper.
>
> >Q5: The assumption of $x_0=0$ should be stated somewhere in the main body of the text explicitly. The assumption requires the system to be initialized at the steady state of the system that the benchmark policy aims to stabilize.
>
> **A5:** We use the condition $x_0=0$ in the proof, and we will add it to the assumptions. Our algorithm does not assume that the system must be initialized at a benchmark-stabilizable steady state; it only requires that the benchmark policy is strongly stable. We appreciate the reviewer's question.
>
>
> **References**
>
> [1] Agarwal, Naman, et al. "Online control with adversarial disturbances." International Conference on Machine Learning. PMLR, 2019.

---

> > ### Comment · Reviewer_H9Pv · 2024-08-09
> >
> > Thank you for your detailed response. Your explanations have clarified most of my concerns, most importantly A1. Below are some clarifications on the points I raised in my original review:
> > * A2: I appreciate your effort to clarify the time indices. However, I am not sure whether removing $h$ would be helpful. My suggestion would have been to, e.g., denote continuous time by $t$ (but keep the use of $h$), discrete time by $k\in\mathbb{N}$, and connecting them by $k=t/h$ to clearly differentiate between continuous time and discrete time steps.
> > * A3: I understood the meaning of the sub- and superscripts of $M$, but I am confused regarding the use of square brackets, e.g., in line 161 (where you use none), line 173 (where the subscript has square brackets) and Algorithm 1 (where the superscript has square brackets in the definition of the DAC policy update class only).
> > * A5: Since the benchmark policy is stable, the corresponding state trajectory will converge to $0$ in the absence of disturbances and to a neighborhood of $0$ for bounded disturbances. Thus, I believe that assuming $x_0=0$ does not just correspond to assuming that the system is initialized at steady state, but at a very specific steady state. However, I also believe that allowing any initialization would only lead to an additional constant cost upper bounding the regret of the transient trajectory, which would be neglected in the $\mathcal{O}$-notation anyway. Therefore, I do not think that this is a major issue.
> >
> > Once again, I would like to thank the authors for their detailed responses. I would like to keep my original score, as it is already relatively high.

---

> > > ### Author Response · Authors · 2024-08-09
> > > **Thank you!**
> > >
> > > We greatly appreciate the reviewer's feedback and suggestions! We will carefully revise these notations to ensure a clearer distinction between discrete and continuous systems. The square brackets in line 173 is a floor function and it is used to simplify notation in subsequent statements. Since our policy parameter updates only once every $m$ steps, the $M$ remains the same within the interval $[im,(i+1)m]$, and we use the floor function to denote the same $M$. The square brackets in Algorithm 1 follow the notation of [1], but we did not realize it overlaps with the previous floor function notation. We will correct this typo in the next version of the paper.
> > >
> > > We once again thank the reviewer for their careful reading of our paper, which helps improve the quality of our work.
> > >
> > > References
> > >
> > > [1] Agarwal, Naman, et al. "Online control with adversarial disturbances." International Conference on Machine Learning. PMLR, 2019.

---

### Official Review · Reviewer_ks8t · 2024-07-13

**Soundness:** 3
**Presentation:** 2
**Contribution:** 3
**Rating:** 6
**Confidence:** 3

**Summary:**

This paper studies the sample complexity of online control for continuous-time linear systems. A novel two-level online algorithm was proposed. Specifically, a sublinear regret is guaranteed. Finally, the authors applied their method to the SAC algorithm and achieves improved simulation results.

**Strengths:**

This paper studies the sample complexity of online control with adversarial disturbances for continuous-time linear systems, providing a complementary perspective to discrete-time linear systems and achieving the same sublinear rate.

**Weaknesses:**

1. There is a gap between the theoretical results and the experimental results. It is not clear how the setting studied in this paper is implemented in the experiments. More discussion is needed.


2. There are several minor writing or notation errors that need to be checked carefully. For example:

line 21 they "has"

line 36 it "reliers"

line 104, missing definition of $\mathcal{K}$

**Questions:**

How the setting studied in this paper is related to the experiments.

**Limitations:**

As mentioned in the paper, the theroical results relies on the access of system model and the loss function is convex. Extentions to model-free methods should be discussed.

---

> ### Author Rebuttal · Authors · 2024-08-07
>
> We express our gratitude to the reviewer for the insightful comments. We address the reviewer's concerns belows:
>
> >Q1: There is a gap between the theoretical results and the experimental results. More discussion is needed.
>
> **A1:** We appreciate the reviewers for raising this issue. On a high level, we first identify the connection between the popular technique known as **domain randomization** and a theoretical framework knwon as **nonstochastic control**. Next, we noticed that existing nonstochastic control algorithms **cannot be readily applied to common domain randomization experiments**. Therefore,  we propose a c**ontinuous time** analysis and highlight two adaptations (skip and stack). We then verify that the technique as well as the analyses indeed can improve upon vanilla policy optimization for domain randomization problems.
>
> We will provide more detailed explanation below. First, we highlight some definitions of stochastic and robust control problem.
>
> In the context of stochastic control, as discussed in [1], the disturbance $w_t$ follows a distribution $\nu$, with the aim being to minimize the expected cost value:
> \begin{equation}
> \min_{\mathcal{A}} \mathbb{E}_{w_t \sim \nu} [J_T(\mathcal{A})]. \tag{Stochastic control}
> \end{equation}
>
> In the realm of robust control, as outlined in [2], the disturbance may adversarially depend on each action taken, leading to the goal of minimizing worst-case cost:
> \begin{equation}
> \min_{u_1} \max_{w_{1: T}} \min_{u_2} \ldots \min_{u_t} \max_{w_T} J_T(\mathcal{A}). \tag{Robust control}
> \end{equation}
>
> We then proceed to **establish the framework for domain randomization**. When training the agent, we select an environment from the distribution $\nu$ in each episode. Each distinct environment has an associated set of disturbances ${w_t}$. However, these disturbances remain fixed once the environment is chosen, unaffected by the agent's interactions within that environment. Given our limited knowledge about the real-world distribution $\nu$, we focus on optimizing the agent's performance within its training scope set $\mathcal{V}$. To this end, we aim to minimize the following cost:
> \begin{equation}
>     \min_{\mathcal{A}} \max_{\nu \in \mathcal{V}} \mathbb{E}_{w \sim \nu} [J_T(\mathcal{A})] . \tag{Domain randomization}
> \end{equation}
>
> From the aforementioned discussion, it becomes clear that the Domain Randomization (DR) setup **diverges significantly from traditional stochastic or robust control**. Firstly, unlike in stochastic frameworks, the randomness of disturbance in DR **only occurs during the initial environment sampling**, rather than at each step of the transition. Secondly, since the system dynamics **do not actively counter the controller**, this setup does not align with robust control principles.
>
>
> So how should we analyze DR from a linear control prospective? Notice that [3] also introduces the concept of non-stochastic control. In this context, the disturbance, while not disclosed to the learner beforehand, remains fixed throughout the episode and does not adaptively respond to the control policy. Our goal is to minimize the cost without prior knowledge of the disturbance:
> \begin{equation}
>     \min_{\mathcal{A}} \max_{w_{1: T}} J_T(\mathcal{A}) . \tag{Non-stochastic control}
> \end{equation}
>
> In this framework, there's **a clear parallel to domain randomization**: fixed yet unknown disturbances in non-stochastic control mirror the unknown training environments in DR. As the agent continually interacts with these environments, it progressively adapts, mirroring the adaptive process seen in domain randomization. Therefore, we propose to **study DR from a non-stochastic control perspective**. As [3] only consider the discrete system, we extent it to continuous system and integrate the two-stage algorithm design idea into the domain randomization experimental environment to see whether it lead to some improvements.
>
> >Q2: There are several minor writing or notation errors that need to be checked carefully.
>
> **A2:** We sincerely appreciate the reviewers pointing out these minor errors, and we will correct them in the next version of the paper.
>
>
> Finally, we thank the reviewer once again for the effort in providing us with valuable and helpful suggestions. We will continue to provide clarifications if the reviewer has any further questions.
>
> **References**
>
> [1] Cohen, Alon, et al. "Online linear quadratic control." International Conference on Machine Learning. PMLR, 2018.
>
> [2] Khalil, I. S., Doyle, J. C., & Glover, K. (1996). Robust and optimal control (Vol. 2). Prentice hall.
>
> [3] Agarwal, Naman, et al. "Online control with adversarial disturbances." International Conference on Machine Learning. PMLR, 2019.

---

### Official Review · Reviewer_2dzb · 2024-07-13

**Soundness:** 3
**Presentation:** 3
**Contribution:** 3
**Rating:** 7
**Confidence:** 4

**Summary:**

The work tackles the challenge of applying simulated controllers to real-world scenarios to manage continuous-time linear systems that face unpredictable disturbances. It introduces a double-layer control approach that balances slow policy updates with quick feedback adjustments. This way it effectively minimizing errors over time. The work is tested on various real world tests in simulation environments and it shows that it significantly boosts the performance of agents, even under uncertain situations.

**Strengths:**

-	Introduction of a new two-level online control strategy that manages both policy learning & feedback control
-	Good theoretical analysis that achieves sublinear regret
-	Validation of theory with practical experiments using domain randomization in various robotic simulation environments

**Weaknesses:**

-	The paper lacks a comprehensive comparative analysis with other state-of-the-art methods.
-	Experiments are conducted in simulated environments with domain randomization. There is limited discussion on how well these results generalize to physical systems outside of controlled simulations

**Questions:**

-	In Section 3.1: the authors assumes access to \cdot{x}/t at each time step. This is a strong assumption, as measuring the state derivative directly is often not possible in practice. Work should discuss implications of this assumption & show it can be extended to more realistic case of only having access to state measurements.
-	Regret defn in section 3.4 compares to the best linear policy in hindsight. Why this is an appropriate comparator class? Why not best non-linear policy? more discussion needed on this choice of benchmark & limitation
-	Assumption 1 is quite strict. What if disturbances may occasionally have large spikes leading to violating these bounds (which happens in practical setting)? include discussion on this case
-	Assumption 2 on the cost function seems to imply the costs are quadratic (due to the D^2 term). But the paper also says the costs are convex, which is a more general class. Clarify precise assumption on the cost function here
-	Definition 1 of a strongly stable linear policy is central to the paper, but the motivation behind this definition is not entirely clear.
-	The challenge of trading off approximation error & OCO regret in section 4 is subtle. More details needed on why this trade-off arises & how this two-level algorithm resolves it.
-	$w_t$ notion is inconsistent. line \sims 100 w_t is treated as a random variable with an unknown distribution. In line 98, it is a specific realization of the disturbance at time t, rather than a random variable.

**Limitations:**

-	Strict assumptions restrict the applicability of finding systems that meet these criteria
-	Method is tailored for non-stochastic disturbances, leaving out scenarios where disturbances might have stochastic elements.

---

> ### Author Rebuttal · Authors · 2024-08-07
>
> We greatly appreciate the reviewer's valuable suggestions. We address the reviewer's questions as follows:
>
> >Q1: The paper lacks comparison with other state-of-the-art methods.
>
> **A1:** We appreciate the reviewer's suggestion, and we will include a discussion of some recent works in the next version of the paper. We only compared our work with [3] in the paper. The papers [4][5] also focus on online control setups with continuous-time stochastic linear systems and unknown dynamics. They achieve $O(\sqrt{T}\log(T))$ regret by different approaches. [4] uses the Thompson sampling algorithm to learn optimal actions. [5] takes a randomized-estimates policy to balance exploration and exploitation.
>
> The main difference between [3][4][5] and our paper is that **they consider stochastic noise of Brownian motion, while the noise in our setup is adversarial**. This makes our analysis completely different from theirs.
>
> >Q2: Experiments are conducted in simulated environments.
>
> **A2:** We admit that we cannot run real-world experiments in a limited time. We believe that applying the algorithm to physical systems is a meaningful future direction.
>
> >Q3: The assumption of the access to $\dot{x}_t$ at each time step is strong.
>
> **A3:** We did not assume access to the derivative of $x_t$ in the paper. As can be seen from Algorithm 1, we only require access to $x_t$. The design of the policy $u_t$ relies on $x_t$ and the estimated values of noise $\hat{w}_t$. The estimation $\hat{w}_t$, as shown in Equation 1, only depends solely on the past values of $x_t$ and $u_t$, not the state derivative.
>
> >Q4: Explain the definition of regret in section 3.4.
>
> **A4:** Our definition of regret is based on [1] and [2], which study online control in discrete systems. In [1], the noise is stochastic, whereas in [2], the noise is adversarial. Lemma 4.3 in Paper 1 proves that under stochastic conditions, the optimal strategy is a strongly stable linear policy. Therefore, regret is measured against the optimal strongly stable policy. Paper 2 continues to use this definition of regret. Consequently, we have adopted their definition.
>
> >Q5: Assumption 1 and 2 are quite strict.
>
> **A5:** Our use of Assumptions 1 and 2 follows those in paper [2] without modification. [2] primarily analyzes discrete systems, and our contribution is **extending their results to continuous systems without additional assumptions**. Furthermore, a function being bounded by a quadratic function does not necessarily imply that it is quadratic. For example, consider the function $C(x,u) = |x|^{1.5}+|u|^{1.5}$, it also satisfies this assumption.
>
> >Q6: The motivation of Definition 1.
>
> **A6:** We appreciate the reviewers' question regarding the motivation behind the definition of "strongly stable." This definition originates from [1] and [2]. Strong-stability is a quantitative version of stability, any stable policy is stronglystable for some $\kappa$ and $\gamma$ (See Lemma B.1 in [1]). Conversely, strong-stability implies stability. A strongly stable policy is a policy that exhibits fast mixing and converges quickly to a steady-state distribution. Thus, **this definition is equivalent to the stable policy**, with the constants $\kappa$ and $\gamma$ introduced primarily for non-asymptotic theoretical analysis to obtain **a more precise** convergence bound.
>
> >Q7: The challenge of trading off approximation error & OCO regret in section 4 is subtle.
>
> **A7:** We appreciate the reviewer for raising this valuable question. Initially, we set out to follow the approach from [2], **where the OCO parameters are updated at each step**. Our regret is primarily composed of three components:  the error caused by discretization $R_1$, the regret of OCO with memory $R_2$ and the difference between the actual cost and the approximate cost $R_3$. The discretization error $R_1$ is $O(hT)$, therefore if we achieve $O(\sqrt{T})$ regret, we must choose $h$ no more than $O(\frac{1}{\sqrt{T}})$.
>
> If we update the OCO with memory parameter at each timestep follow the method in [2], we will incur the regret of OCO with memory $R_2 = O(H^{2.5}\sqrt{T})$. The difference between the actual cost and the approximate cost $R_3 = O(T(1-h\gamma)^{H})$. To achieve sublinear regret for the third term, we must choose $H = O(\frac{\log T}{h\gamma})$, but since $h$ is no more than $O(\frac{1}{\sqrt{T}})$, $H$ will be larger than $\Theta(\sqrt{T})$, therefore the second term $R_2$ will definitely exceed $O(\sqrt{T})$.
>
> Therefore, we adjusted the frequency of updating the OCO parameters by **using a two-level approach and update the OCO parameters once in every $m$ steps**. This will incur the third term $R_3 = O(T(1-h\gamma)^{Hm})$ but keep the OCO with memory regret $R_2 = O(H^{2.5}\sqrt{T})$, so we can choose $H = O(\frac{\log T}{\gamma})$ and $m=O(\frac{1}{h})$. Then the term of $R_2$ is $O(\sqrt{T}\log T)$ and we achieve the same regret compare with the discrete system.
>
> Once again, we thank the reviewer for the constructive comments.
>
> **References**
>
> [1] Cohen, Alon, et al. "Online linear quadratic control." International Conference on Machine Learning. PMLR, 2018.
>
> [2] Agarwal, Naman, et al. "Online control with adversarial disturbances." International Conference on Machine Learning. PMLR, 2019.
>
> [3] Basei, Matteo, et al. "Logarithmic regret for episodic continuous-time linear-quadratic reinforcement learning over a finite-time horizon." Journal of Machine Learning Research 23.178 (2022): 1-34.
>
> [4] Shirani Faradonbeh, Mohamad Kazem, Mohamad Sadegh Shirani Faradonbeh, and Mohsen Bayati. "Thompson sampling efficiently learns to control diffusion processes." Advances in Neural Information Processing Systems 35 (2022): 3871-3884.
>
> [5] Faradonbeh, Mohamad Kazem Shirani, and Mohamad Sadegh Shirani Faradonbeh. "Online Reinforcement Learning in Stochastic Continuous-Time Systems." The Thirty Sixth Annual Conference on Learning Theory. PMLR, 2023.

---

> > ### Comment · Reviewer_2dzb · 2024-08-14
> >
> > Thank you for the detailed response. Some of the answers have helped clarify the questions raised in my review. I have no further questions at this point. I hope the authors can leverage the rebuttal answers (specifically A1, A2 and A6) in the subsequent version of this manuscript.

---

> > > ### Author Response · Authors · 2024-08-14
> > > **Thank you**
> > >
> > > We thank the reviewer for acknowledging our work. We will incorporate the reviewer's valuable suggestions in the next version of our paper.

---

### Decision · Program_Chairs · 2024-09-25

**Decision:**

Accept (poster)

**Comment:**

Summary: Most of the existing works on online control have been on discrete-time linear systems, except for a handful of works that consider stochastic noise. This paper studies an online control problem with adversarial disturbances for a continuous time linear system. This paper proposes a two-level online controller to solve this problem. The higher-level controller symbolizes the policy learning process and updates the policy at a low frequency to minimize regret. The lower-level controller delivers high-frequency feedback control input to reduce discretization error. The paper shows that the proposed algorithm achieves sublinear regret in the face of non-stochastic disturbances. The performance of the proposed algorithm is illustrated using simulation experiments in the Mujoco setting.

We received four expert reviews, with the scores 7, 6, 6, 4, and the average score is 5.75. Reviewers are generally positive about the technical contribution and algorithmic novelty. They are also satisfied with the simulation experiments. One reviewer has expressed concerns about two aspects of technical proofs, in particular about the difficulty of analysis and the novel contribution. I believe that the authors have given a detailed and satisfactory answer to this comment.

The reviewers have given many critical feedback on improving the quality of the paper. The authors have acknowledged them and agreed to incorporate these changes in their revision. Please make sure that you update your paper based on these comments while preparing the final submission.